

# Field data to benchmark the carbon-cycle models for tropical forests

Deborah A. Clark[1], Shinichi Asao[2,3], Rosie Fisher[4], Sasha Reed[5], Peter B. Reich[6,7], Michael
G. Ryan[2,8], Tana E. Wood[9], Xiaojuan Yang[10]

[1] Department of Biology, University of Missouri-St. Louis, Saint Louis 63121, MO, USA

[2] Natural Resource Ecology Laboratory, Colorado State University, Fort Collins, CO 80523-1499, USA

[3] ARC Centre of Excellence in Plant Energy Biology, Research School of Biology, The Australian National University,
Canberra, ACT 0200, Australia

[4] Terrestrial Sciences Section, Climate and Global Dynamics, National Center for Atmospheric Research, Boulder, CO
80301, USA

[5] US Geological Survey, Southwest Biological Science Center, Moab, UT 84532, USA

[6] Department of Forest Resources, University of Minnesota, St. Paul, MN 55108, USA

[7] Hawkesbury Institute for the Environment, Western Sydney University, Penrith, NSW 2751, Australia

[8] Rocky Mountain Research Station, USDA Forest Service, Fort Collins, CO. 80526 USA

[9] International Institute of Tropical Forestry, USDA Forest Service, Rio Piedras, PR 00926, USA

[10] Oak Ridge National Laboratory, Climate Change Science Institute and Environmental Sciences Division, Oak Ridge, TN
37831-6335, USA

*Correspondence to:* Deborah A. Clark (deborahanneclark@gmail.com)

**Abstract.** For more accurate projections of both the global carbon (C) cycle and the changing climate, a critical current need
is to improve the representation of tropical forests in Earth system models. Tropical forests exchange more C, energy, and
water with the atmosphere than any other class of land ecosystems. Further, tropical-forest C cycling is likely responding to
the rapid global warming, intensifying water-stress, and increasing atmospheric $CO_2$ levels. Projections of the future C
balance of the tropics vary widely among global models. A current effort of the modeling community, ILAMB (the
International Land Model Benchmarking Project), is to compile robust observations that can be used to improve the accuracy
and realism of the land models for all major biomes. Our goal with this paper is to identify field observations of tropical-
forest ecosystem C stocks and fluxes that can support this effort. We propose criteria for reference-level field data from this
biome and present a set of documented examples from old-growth lowland tropical forests. We offer these as a starting point
towards the goal of a regularly updated consensus set of benchmark field observations of C-cycling in tropical forests.



## 1 Introduction

*"*The near-future research effort should be on development of a set of widely acceptable benchmarks that can be used to objectively, effectively, and reliably evaluate fundamental properties of land models to improve their prediction performance skills." (Luo et al., 2012)

Improved modeling of tropical-forest carbon (C) cycling is urgently needed for projecting future climate and for guiding global policy concerning greenhouse gases. Tropical forests are major players in the global C cycle. These ecosystems store an estimated 25% of terrestrial C stocks (Bonan et al., 2008), they exchange vast quantities of carbon dioxide ($CO_2$) with the atmosphere (Beer et al., 2010), and their C cycling is climatically sensitive (Clark et al., 2003; Balser & Wixon, 2009; Wood

et al., 2012; Clark et al., 2013). Atmospheric inverse models indicate that temperature-linked changes in the annual C balance of the land tropics during recent decades (higher tropical emissions in hotter years) have largely driven the marked inter-year changes in the growth rate of atmospheric $CO_2$ ($[CO_2]$), after factoring out fossil-fuel emissions (Ciais et al., 2013; also Anderegg et al., 2015).

In addition to the on-going effects of deforestation and fires, climate change is likely to magnify the biome's large

role in global C-cycling. Tropical forests are being rapidly moved into new climate territory (Wright et al., 2009). One Earth system model (ESM) has projected that, during the next 25 years, up to 70% of seasons in the tropics will be hotter than all the corresponding seasons before 2000 (Diffenbaugh and Scherer, 2011). While future tropical rainfall regimes remain highly uncertain (Collins et al., 2013), it is clear that warming is also progressively increasing relative air dryness (Vapor Pressure Deficit, VPD; Sherwood and Fu, 2014), placing another downward pressure on tropical-forest productivity (Clark et

al., 2013). Although some ecophysiological theory indicates that increasing $[CO_2]$ could mitigate these stresses (Lloyd and Farquhar, 2008), such "$CO_2$ fertilization" for tropical forests is expected to be constrained by widespread nutrient limitation (Townsend et al., 2011; Goll et al.. 2012; Wieder et al., 2015) and is also likely to be offset by the increasingly negative effects of climate change across the tropics (Wood et al., 2012; Clark et al., 2013; Smith et al., 2016). The net effect of all these environmental factors will strongly affect how this biome contributes to, or detracts from, the land C sink in coming

decades, with large consequences for the pace of global warming.

Projecting the future integrated effects of climatic and atmospheric change on tropical forest C cycling can only be approached through process-based modelling. Current models, however, strongly disagree among themselves with respect to tropical forests, thus producing major uncertainties for global diagnosis and planning. While some coupled ESMs indicate increasing net C uptake by the land tropics through this century, others project a progressive decline in the net flux, with the

spanned difference approaching 7 Pg C yr$^{-1}$ by 2100 (Fig. 1). Multiple studies (Delbart et al., 2010; Negrón-Juárez et al., 2015) have reported large mis-matches between spatially-referenced ground observations (tropical-forest aboveground biomass, woody productivity, tree mortality) and the corresponding outputs from ESMs in the CMIP5 studies (Coupled Model Intercomparison Project, phase 5). A further indication of unresolved issues for modeling this biome is that nine of





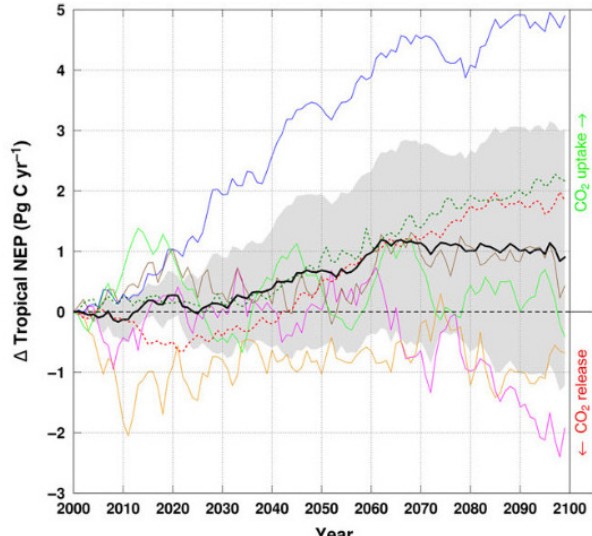

**Figure 1.** Divergent projections of the changes in tropical Net Ecosystem Production through this century from seven of the CMIP5 climate models (adopted with permission from Cavaleri et al., 2015 [© 2015 John Wiley & Sons Ltd]).

ten C-cycle models failed to simulate the climatic responses of the global land C sink through 1980-2009 as inferred from the atmospheric data (most models overestimated the land sink's sensitivity to rainfall and/or underestimated its sensitivity to temperature; Fig. 6.17 in Ciais et al., 2013).

To improve current global C cycle models, a community-wide effort – ILAMB (The International Land Model Benchmarking project) – seeks to identify robust observations from each biome (hereafter, "benchmark data") that can serve to guide model structure and to enable standardized tests of the models (Luo et al., 2012). Our goal with this paper is to contribute to the ILAMB effort by identifying such reference-level field observations from tropical forests to guide the models for this biome. We restrict our focus to the most extensive and most C-rich  sector of the biome (Raich et al., 2006): old-growth forests in the tropical lowlands (elevations < 500 m). Given the large footprint of global models (e.g. km-scale), we additionally focus specifically on larger-scale, landscape-level ecosystem fluxes and pools rather than on data required for refining functions and relationships within models. While we recognize the need to incorporate nutrient cycling into global models, we limit our focus to carbon, although the criteria used here could be applied to nutrient fluxes and pools as well. We first propose criteria for identifying benchmark-level field observations from these forests. We then review the current availability of such data and present a set of documented examples. We offer these ideas and examples as a starting point towards the goal of a constantly updated consensus set of benchmark field observations for the tropical-forest biome.



## 2 Types of model-data interactions

Field observations from tropical forests can help develop and validate models in multiple ways. First, for each C-cycle model, the prescribed and diagnostic ecosystem metrics for the biome should be comparable to the relevant field data. For instance, do the modelled Leaf Area Index (LAI), aboveground live biomass, and aboveground wood production fall within

the 95% confidence limits of the observations from tropical forests? Do relationships among stocks and fluxes match the relationships found among the field observations? Such questions can be posed at the biome level or for specific tropical regions, depending on a model's spatial resolution and the available data. The pattern of spatial variation in model outputs for different tropical-forest regions can be tested against the field observations (e.g., Negrón-Juárez et al., 2015). Observations from tropical-forest field sites can also be used to evaluate the results from site-specific model experiments for the years

spanned by those field studies. Do the modeled C stocks and ecosystem responses and their interannual variation approximate the observations for the corresponding time period? For all these uses, multiple issues arise for selecting and using appropriate field data, and we discuss these individually in the following sections.

### 2.1 Comparing apples to apples

A general consideration for model-data interactions is that the field studies to date in tropical forests have addressed only

some of the forest attributes and processes involved in C cycling. Considerable uncertainty is introduced when models are compared to hybrid C-cycle estimates that are only partially based on field observations, as in Figure 2. In that case, the first-cut estimates of Malhi et al. (2009) were derived by combining the available field observations for some C-cycle aspects

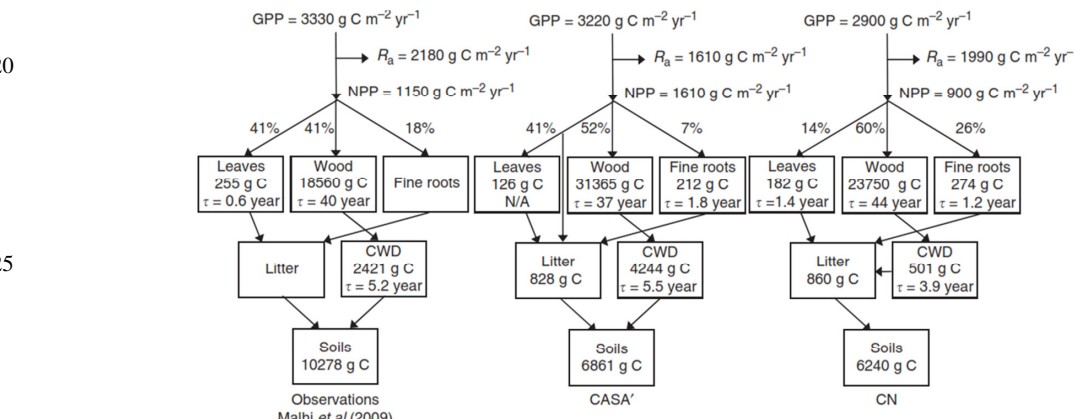

**Figure 2.** A comparison of CASA and CN model outputs to estimates derived by combining the limited field data with estimates

of unmeasured components (from Randerson et al., 2009, with permission [© 2009 Blackwell Publishing Ltd]).



with unverifiable estimates for unmeasured components such as daytime leaf respiration and coarse-root biomass. Other aspects that were omitted may be important in most tropical forests; these include the large $CO_2$ flux from canopy-level branches (Cavaleri et al., 2006) and the summed belowground C exports to mycorrhizae and root exudates. While there can be considerable heuristic value in partially-biometric estimates for C stocks and fluxes, such as those of Malhi et al. (2009),
they do not provide direct observational standards for the models. The most meaningful comparisons of models with field data will be for those specific pools and fluxes that were assessed in the field.

The other side of the "apples to apples" issue is that, for data-model comparisons, many C-cycle models may require development to include or output those specific ecosystem attributes that have been field-quantified in tropical forests (e.g., aboveground wood production, leaf litterfall). Similarly, the land-surface models may need to be re-structured
to better represent properties where only part of the system state can actually be observed (e.g., predicting surface-soil organic C [SOC], rather than total-column SOC; c.f., Koven et al., 2013).

### 3 Criteria for benchmark field data from tropical forests

### 3.1 Direct field measurements

As discussed above, some reported observations of C-cycle attributes are based partly on direct measurements and partly on
extrapolation. An example would be total fine-root production as estimated by extrapolating surface-soil measurements to the unstudied deeper soil layers (e.g., Doughty et al., 2013). Similarly, the tower-based eddy-covariance technique measures forest-level Net Ecosystem Exchange (NEE) of carbon dioxide ($CO_2$). Because this technique does not measure the two component fluxes of NEE, Gross Primary Productivity (GPP) and Ecosystem Respiration ($R_{eco}$), modeling and assumed physiological responses have been used to infer those two fluxes from NEE (Wehr et al., 2016). As recently argued by
Negrón-Juárez et al. (2015), the most meaningful model-data comparisons will be those based as closely as possible on the actual field measurements (i.e., surface-soil fine-root production and NEE, respectively, in the above examples). Because the current field techniques all have clear limitations (Clark et al., 2001a; Cleveland et al., 2015), such observation benchmarks also need to be explicitly associated with the specific method used. If a superior method emerges, those benchmarks would need updating.

### 3.2 Landscape-scale data

"Field measurements can be comparable to the predictions of global NPP models (and could be eventually used for parameterizing them) only when they are collected by a systematic stratified design, and are therefore representative of the given region." (Simova and Storch, 2016)



"... extrapolations and predictions of forest properties based on sparsely and/or nonrandomly distributed field plots are no longer acceptable for understanding tropical forests in regional or global carbon cycles." (Marvin et al., 2014)

"A single plot corresponds to one sample of the forest, and it is unlikely to represent the whole landscape-scale environmental variability." (Chave et al., 2004)

Many key features of C cycling (e.g., C stocks, LAI, productivity) vary within each tropical forest due to the local-scale variation in disturbance histories, edaphic conditions (slope, fertility) and floristics. Indeed, in landscapes that can support

hundreds of tree species per hectare (Losos and Leigh, 2004), the potential for small-scale variability in plant properties, soil characteristics and thus C-cycle attributes is very high. For example, among 18 0.5-ha plots distributed across a Costa Rican old-growth forest, estimated aboveground wood production varied 2-fold (Clark et al., 2013) and the large mortality-driven biomass losses occurred in only a few of the 18 plots (Clark, 2004).

Most land surface models attempt to predict landscape-scale fluxes and pools. Field studies should therefore

provide distributed measurements that span the within-landscape variability. When a forest is instead sampled in only one or two small ($\leq$ 1 ha) plots, as is the case for most sites covered by two current plot networks (RAINFOR in the Amazon, Brienen et al., 2015; AFRITRON in Africa, Lewis et al., 2009), the observations may be unrepresentative of average conditions in those forests. Using remote-sensing over Peruvian tropical forests, Marvin et al. (2014) found that the structural attributes of individual small study plots significantly differed from the landscape-level mean attributes of each sampled

forest.

For typical land surface models, which operate at a scale of 0.5 degrees or larger, benchmark field observations would ideally be based on field measurements distributed over those extremely large areas. Due to both cost and the challenging logistics, however, no field study of ecosystem-level C-cycling has covered such a huge area of tropical forest. Current consensus (e.g., Chave et al., 2004; Rutishauser et al., 2010, Chambers et al., 2013; Marvin et al., 2014) favors two

compromise approaches to representative sampling of a tropical-forest landscape for such studies: 1) measurements over a set of small plots that aggregate to at least 5-10 ha and are distributed to span the important heterogeneity of the studied landscape (e.g., de Castilho et al., 2010; Rutishauser et al., 2010; Clark et al., 2013); or 2) measurements covering a very large plot, such as the 50-ha plots of the Center for Tropical Field Science (CTFS; Anderson-Teixeira et al., 2015). While these prescriptions do not achieve sampling at the scales treated in many ESM's, these compromise "landscape-scale"

sampling approaches can be used to determine the ranges and means of C stocks and fluxes at the mesoscale (e.g., 50-2000 ha).

One class of models contrasts with the ESM's in explicitly representing the small-scale within-landscape heterogeneity caused by the patchwork of disturbance-recovery phases observed in the real world. An example is the Ecosystem Demography model (Moorcroft et al., 2001; Medvigy et al., 2009; Fisher et al., 2015), designed in part to capture



the variation between recently disturbed and old-growth forests. With those models the smaller scale observations, such as those from individual hectares, can be usefully compared directly to the model output.

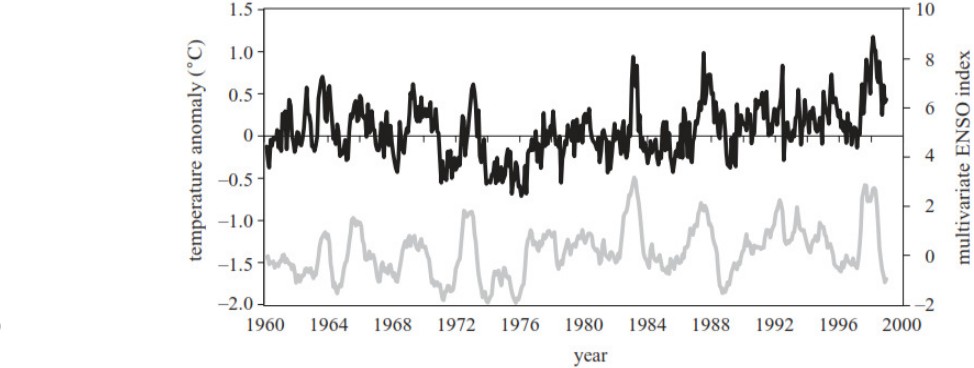

**Figure 3.** Anomalies of pan-tropical mean temperature (black) and the ENSO multivariate index (grey) compared to the period 1960-1990. (from Malhi and Wright, 2004; by permission of the Royal Society).

### 3.3 Long data series

Key outputs from the global models concern the long-term trends in C-cycle attributes in each biome due to both climate

change and increasing atmospheric [$CO_2$]. Field-based reference benchmarks concerning either directional trends through time or the climatic/[$CO_2$] sensitivities of forest C cycling are needed to evaluate this aspect of model outputs. Such observational benchmarks need to be based on long data series. A two-sample comparison, then vs. now (e.g., Lewis et al., 2004), can be consistent with an hypothesized or modelled long-term trend but is insufficient to demonstrate or quantify it. With random draws of two observations from a time series that has no underlying significant temporal trend, on average in

half the cases the second observation will be greater/(less than) the first. As demonstrated by Hall et al. (1998; also Clark and Clark, 2011), for the many tropical-forest processes and attributes that vary substantially among years, short data series are insufficient for reliable detection of long-term declines or increases.

When a long data series does exist for a given C-cycle attribute or process, climatic and/or [$CO_2$] sensitivities of that aspect of forest C cycling can be quantified by statistically relating the observations to the changes in the environmental

drivers. The interannual variation in tropical climatic conditions (Fig. 3) greatly aids such analyses. Valid climatic/[$CO_2$] relationships of C-cycle attributes will increase in statistical significance as more yearly points are added (see Table 3 in Clark et al., 2013). Too-short data series, however, can miss the underlying climatic/[$CO_2$] responses or suggest spurious ones. For annual wood-production in one tropical forest, in a retrospective analysis based on progressively shorter segments of a 24-yr record (Fig. 4), many series of <10 annual re-measurements missed the highly significant negative temperature





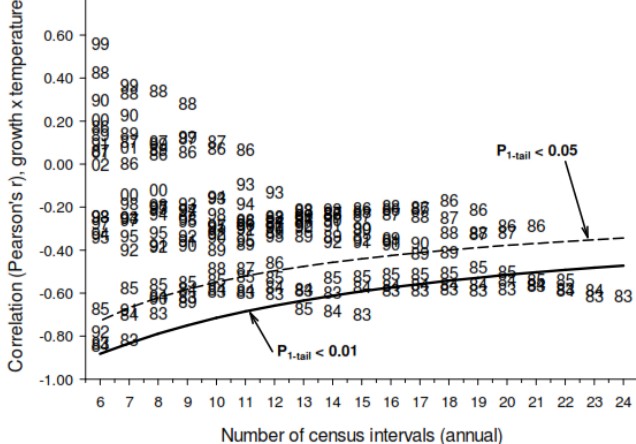

**Figure 4.** Effect of length of data series on the correlation of tree growth with minimum temperatures at La Selva, Costa Rica. Data labels: Year 1 of each segment of the series (from Clark and Clark, 2011; with permission from the Association for Tropical Biology and Conservation).

response that was shown by the full record; some 6-yr series in fact suggested the opposite, likely due to uncontrolled-for variation in other climatic drivers. Ideally, modeling analyses should aim to capture the dominant causes of this inter-annual

variability, where they are non-random. Again, apple-to-apple comparison is critical, looking at the results in the context of local conditions and meteorology, rather than abstracting to larger scales.

### 3.4 Supporting information

For model-data fusion, benchmark field data should be accompanied by several classes of supporting information.

Geographic coordinates of the study site are required for spatially-explicit model tests. Site elevation (m above sea level) locates the finding along the lowland-montane continuum of tropical forests. Given the likelihood of interannual and directional changes in forest C cycling, the year(s) of each study (often also the months) is critical information. Other key specifications include the area sampled, details of the field methods used, and the citation of the study. The web location of the actual data should also be part of each benchmark listing; although this last specification cannot yet be fulfilled for most

tropical-forest field data, changes now underway in publication requirements may soon make this a realistic addition to the data base design.

Ideally, model runs should be set up for individual "testbed" sites, to best allow consideration of site-specific circumstances. Where these types of model-data fusion are planned, a much larger set of auxiliary data is potentially useful,





including high-resolution local meteorological data, soil physical properties (texture, depth), and vegetation properties relevant to the question being posed.

## 4 Benchmark field data from lowland old-growth tropical forests

Using the above criteria (direct field measurements, landscape-scale sampling, sufficiently long data series), we have
extracted from the literature examples of robust ecosystem-level field observations of C cycling in these forests. Not surprisingly we found important data gaps. We also identified significant methods issues for field-quantifying C-cycle attributes. As discussed below, while some of these issues affect C-cycle studies in all forest types, others are particular to tropical-forest conditions. In the following sections, for each C-cycle attribute we review the state of the existing field data and present documented examples of robust field observations, when available. Two areas are specified in the example
tables: the summed area of the actual measurements (e.g., cores, traps), and the total area of the forest over which the measurements were distributed ("Total study area": the area of a polygon encompassing all measurements). Table 14 provides core information on each study site in the preceding data tables.

Table 1 provides a capsule summary of our findings, which are detailed in the following sections. As illustrated in the table, C-cycle attributes vary across space and/or time. Model predictions typically are for a single state in a given place
and time. Increasingly, however, model predictions are made across a range of parameters (Zaehle et al., 2005; Fischer et al., 2011), initial conditions (Lombardozzi et al., 2014), driving data (Fox et al., 2009; Viskari et al., 2015) and structural variations (Fisher et al., 2015; Medlyn et al., 2015), resulting in ranges of predictions that can be compared against observations which themselves are known to have errors. Therefore, it is not strictly necessary that observational benchmarks have very low confidence ranges, but it is necessary to document that range of observations and the natural
variability that the observations span.





**Table 1.** Summary of the characteristics of field observations of ecosystem C-cycling in lowland old-growth tropical forests, from the example data presented in this paper (in the tables or text, or footnotes here). "n.d." – no benchmark field observations yet identified from this biome. Attribute abbreviations are defined in the text.

| C-cycle attribute | Range of forest-level means | Min/Max a good indicator of lower or upper bound: | Within-site variation example sites: Ha to ha | Yr to yr | Salient issues for attribute in tropical forests |
|---|---|---|---|---|---|
| LAI (full canopy) | 4–6 | (both bounds)[1] | | | The two direct harvests indicate max. LAI ca. 6; optical methods underestimate |
| *Ecosystem C stocks:* | | | | | |
| Total C stocks | n.d. | | | | Unquantified components could sum to >50% of total C stocks |
| Aboveground live biomass | 161–497 Mg ha⁻¹ | | 2.4 | 1.0–1.06 | Estimates are typically for larger stems and are based on unverified allometry |
| Coarse roots | n.d. | | | | No stand-level field observations |
| Fine roots | >0.5–8.0 Mg ha⁻¹ | lower bound | 1.2–1.4 | 3.75[2] | Data are confined to surface soil |
| Coarse woody debris | 20–96 Mg ha⁻¹ | | | | Few landscape-scale data; highly variable in space and time |
| Soil organic C | >213–373 Mg C ha⁻¹ | lower bound | 1.75[3] | | Almost never quantified to maximum soil depth or through time |
| *Ecosystem C fluxes:* | | | | | |
| Annual NEE of CO₂ | n.d. | | | | Issues for eddy-flux in tropical forests make annual NEE problematic (see below) |
| GPP | n.d. | | | | Biometric omissions could sum to >50%; GPP is not measured by eddy-flux |
| Rₐ, Rₕ | n.d. | | | | Field observations in tropical forests are incomplete and ambiguous (see below) |
| Total NPP | n.d. | | | | Biometric omissions could sum to >50%. Total NPP is not measured by eddy-flux |
| Aboveground wood production | 3.7–8.7 Mg ha⁻¹ yr⁻¹ | | 1.4–2.1[4] | 1.4[5] | Usually only larger stems (≥ 10–35 cm diameter); based on unverified allometry |
| Mortality biomass loss | 5.0–8.0 Mg ha⁻¹ yr⁻¹ | | 2.5–15.0 | 2.9 | Marked spatiotemporal variation; based on unverified allometry |
| Leaf production | n.d. | | | | No stand-level observations |
| Leaf litterfall | > 5.7–6.8 Mg ha⁻¹ yr⁻¹ | lower bound | 1.6–2.2[4] | 1.2[5] | Always an underestimate; excludes pre-collection losses (see Table 7) |
| Twig litterfall | >0.9–2.5 Mg ha⁻¹ yr⁻¹ | lower bound | 2.7–8.7[4] | 1.5[5] | Always an underestimate; excludes pre-collection losses (see Table 7) |
| Reproductive litterfall | >0.4–1.3 Mg ha⁻¹ yr⁻¹ | lower bound | 2.5–6.4[4] | 1.4[5] | Always underestimate; excludes consumption |
| Fine-root production | >0.7–3.4 Mg ha⁻¹ yr⁻¹ | lower bound | | | Only in surface soil; significant methods issues |
| Plant C exports to symbionts | n.d. | | | | Unquantified in tropical forest; possibly a large and increasing fraction of NPP |
| Root exudates | n.d. | | | | Unquantified in tropical forest; possibly a large and increasing fraction of NPP |
| Volatile organics production | n.d. | | | | Unquantified in tropical forest; likely a small but increasing fraction of NPP |

[1] minimum from indirect methods likely a good indicator of lower bound of LAI; 6 is a reasonable upper bound (but based on only 2 harvest studies)

[2] 8-yr max and 8-yr min of stocks of live fine roots (< 2 mm, 0–50cm depth) on old oxisols, LS site (Espeleta and Clark, 2007)

[3] ratio, soil organic carbon to 3 or 4 m depth in old oxisols vs. in younger oxisols, LS site (Table 6; Veldkamp et al., 2003)

[4] range of ratios of max to min values from 18 0.5-ha plots in each of 12 successive years, LS site (Clark et al., 2013)

[5] ratio between 12-yr max and 12-yr min of yearly means of 18 0.5-ha plots, LS site (Clark et al., 2013)



### 4.1 Leaf Area Index (LAI)

Field observations for this often prognostic model parameter are methods-dependent and typically underestimate (see Table 2). Forest-level LAI can be assessed in the field directly, if laboriously, through replicated leaf harvests from the canopy top to the forest floor. To date, however, only one study (Clark et al., 2008) has directly assessed it this way in a tropical forest

5 (**LS** site, Table 2). Harvested LAI at their 55 4.6-m$^2$ stratified-random sampling points across that forest ranged from 1.2 to 12.9, reflecting the spatial heterogeneity of tropical-forest LAI and thus the need for distributed replicate sampling. Parallel estimates were also made with the two indirect techniques (LAI-2000, hemispherical photographs) that are the standard current approaches for estimating LAI in the field. Both indirect methods were found to saturate in sites of overhead LAI > 6, resulting in 12-38% underestimates of the direct harvest data, depending on the adjustments made for wood and/or leaf-

10 clumping (Olivas et al., 2013). In one other study involving direct harvest of all leaves from the forest floor to the canopy top in a 20 m x 20 m plot (McWilliam et al., 1993; see Table 2), the value obtained was similarly at the high end of tropical-forest LAI observations.

**Table 2.** LAI observations in lowland old-growth tropical forests.

| LAI | Method | Area (ha) | Region | Site Code | Source of data | Method details |
|---|---|---|---|---|---|---|
| 6.00 | Direct harvests | 500 | C. AMER | LS | Clark, D.B. et al., 2008 | floor to canopy top leaf harvests, 55 points across 500 ha |
| 5.10 | LAI-2000 | 500 | " | " | Olivas et al., 2013 | at >1 m ht at 55 direct-harvest sites |
| 4.9-6.0 | Hemisph. photos | 500 | " | " | Olivas et al., 2013 | at >1 m ht, 55 harvest sites; WinSCANOPY output types |
| 3.90 | Hemisph. photos | 500 | " | " | Olivas et al., 2013 | at >1 m ht, 55 direct-harvest sites; Gap Light Analyzer |
| 2.7-4.85 | Hemisph. photos | 9 | " | " | Loescher et al, 2003 | at >1 m ht; N=6 in each of 18 plots; 3 wet/dry seasons |
| 5.70 | Direct harvests | 0.04 | AMAZON | MAN-McW | McWilliam et al., 1993 | harvested 4 10x10m contiguous sections of forest |
| 4.45 | Hemisph. photos | 2 | AMAZON | AGP-01,02 | Jiménez et al., 2014 | at 1 m ht; N=26/ha, unknown number of visits; Hemiview |
| 4.25 | Hemisph. photos | 1 | AMAZON | ZAR-01 | Jiménez et al., 2014 | at 1 m ht; N=26/ha, unknown number of visits; Hemiview |
| 5.58 | Hemisph. photos | 1 | AMAZON | MAN-K34 | Marthews et al, 2012 | at 1 m ht; no details ("unpubl., S. Patiño") |
| 5.25 | Hemisph. photos | 2 | AMAZON | CAX-06 | Marthews et al, 2012 | at 1 m ht; no details ("unpubl., S. Patiño") |
| 5.30 | Hemisph. photos | 1 | AMAZON | CAX-CTL | Metcalfe et al., 2010 | at 1 m ht, 25 points in 1 ha, 1 date; Hemiview |
| 4.3-5.7 | LAI-2000 | 1 | AMAZON | CAX-CTL | Metcalfe et al., 2010 | 100 points, unknown height, 5 dates |
| 5.03 | LAI-2000 | 3.1 | AMAZON | TAP-KM67 | Malhado et al., 2009 | monthly over 1 yr; range of monthly values 4.8-5.2 |
| 4.8-5.1 | LAI-2000 | 1.5 | AMAZON | TAP-A1,A4 | Aragão et al., 2005 | 2 forests, 3 0.25 ha plots ea, 25 points per plot, at unk. ht. |

### 4.2 Ecosystem C stocks

The total ecosystem C inventory has not been quantified in any tropical forest. Field-quantifying this C-cycle attribute

25 would be challenging for any forest type. Impediments in tropical forests include difficulty of access, harsh climatic conditions, marked within-forest variation, and the complex forest structure. Most frequently estimated in this biome is the aboveground biomass of the larger live woody stems. Components of live biomass that are as yet unquantified at the stand level in these forests include: coarse roots; subsurface fine roots; epiphytes; hemiepiphytes; and understory plants. Coarse woody debris is rarely estimated. When soil organic carbon (SOC) is assessed, sampling is nearly always confined to the



surface soil. For modeling, the available data from tropical forests provide a lower bound on total C stocks. These data are most valuable, however, at the level of individual components.

***Live aboveground biomass.*** All field observations of live aboveground biomass in tropical (and non-tropical) forests are indirect, unvalidated estimates for just the larger stems (EAB - Estimated Aboveground Biomass). For multiple

reasons (see below), it remains unclear how the existing EAB values for this biome can best serve the models.

To derive EAB, all live stems in a stand above some diameter limit (usually 10 cm) are measured for diameter (rarely also height). Each stem's aboveground biomass is then estimated using an allometric relationship between biomass and diameter (/height) that was derived by harvesting and weighing individual trees at another site(s). This approach raises the issue of "...misplaced concreteness" with respect to forest biomass estimates (Clark and Kellner, 2012). Different

allometric equations can produce starkly different values of EAB from the same set of stem measurements; this is illustrated in Table 3 by the range of the five estimates (242-428 Mg/ha) produced by different allometries but from the same 1992 set of tree-diameter inventory data at the **NOU-PP** site. To determine which, if any, of such estimates is accurate for a given landscape would have required follow-up structured harvests at the site to test the applicability of a given allometric relation to that forest (Clark and Kellner, 2012). Because as yet no such validation has been carried out in a tropical forest, all EAB

values for this biome are highly uncertain at the site level. While the range of these estimates is the only available guidance for upper and lower bounds for this biome, the accuracy of this range is also unknowable. Given these uncertainties, it will be important to maintain the actual field data (e.g., diameter and taxonomy of all stems) in a publically-accessible archive, so that users could apply alternative allometries or estimation methods in the future.

For testing models against field observations of tropical-forest biomass (c.f., Cleveland et al., 2015), a separate

important issue is the within-forest spatial heterogeneity of EAB. For example, within a 10-ha area of French Guianan forest where EAB averaged 301 Mg ha$^{-1}$ (NOU-GP in Table 3) the range of the estimates for individual hectares was 230-416 Mg ha$^{-1}$ (Chave et al., 2001). A similarly large range among individual hectares was also found within the 50-ha plot on Barro Colorado I., Panama (180-440 Mg ha$^{-1}$; Chave et al., 2003). Due to this local-scale variation, landscape-scale biomass observations would be required for most types of model-data fusion (except in the case of demographic models [Hurtt et al.,

2004], which explictly incorporate this spatial heterogeneity).

Many models, particularly those that simulate forest demographics, use allometric equations to relate stem diameter to biomass. They also typically use estimated production of woody biomass to calculate diameter increments. In such cases, comparisons of both biomass and diameter increment for the same forest are therefore only sensible if the same allometric scaling is used. Again, detailed knowledge both of the data products (including EAB) and of model structures is critical.

Current ILAMB benchmarks for tropical regions include maps of aboveground biomass across the biome based on



remote-sensing products (e.g., Saatchi et al., 2011; Baccini et al., 2012). Large divergences between these maps (Mitchard et al., 2014) highlight the unresolved uncertainties due to methods issues for both the remote-sensed data and the field observations (e.g., unvalidated allometries, landscape-scale samples vs. a single 1-ha plot).

**Table 3.** Landscape-scale estimates of aboveground biomass in lowland old-growth tropical forests.
Estimates are based on diameters of all live stems in 9-72 ha per site. Lianas (+ or -): lianas included in biomass estimate?

| EAB (Mg ha$^{-1}$) | Measured area (ha) | Total study area (ha) | Region | Site code | Citation | Min. diam. (cm) | Lianas | Allometry used | Year(s) |
|---|---|---|---|---|---|---|---|---|---|
| 242 | 12 | 12 | GUIANAS | NOU-PP | Chave et al., 2001 | 10 | - | Brown, 1997 (trop. wet) | 1992 |
| 317 | " | " | " | " | Chave et al., 2001 | 10 | - | Chave et al., 2001 | 1992 |
| 428 | " | " | " | " | Chave et al., 2001 | 10 | - | Lescure et al., 1983 | 1992 |
| 376 | " | " | " | " | Chave et al., 2008b | 10 | - | Chave et al., 2005 | 1992 |
| 381 | " | " | " | " | Chave et al., 2008b | 10 | + | varied with plant type | 1992 |
| 398 | " | " | " | " | Chave et al., 2008b | 10 | - | Chave et al., 2005 | 2000-02 |
| 403 | " | " | " | " | Chave et al., 2008b | 10 | + | varied with plant type | 2000-02 |
| 301 | 10 | 10 | GUIANAS | NOU-GP | Chave et al., 2001 | 10 | - | Chave et al., 2001 | 1992-94 |
| 356 | " | " | " | " | Chave et al., 2008b | 10 | - | Chave et al., 2005 | 1992-94 |
| 366 | " | " | " | " | Chave et al., 2008b | 10 | + | varied with plant type | 1992-94 |
| 356 | " | " | " | " | Chave et al., 2008b | 10 | - | Chave et al., 2005 | 2000-02 |
| 366 | " | " | " | " | Chave et al., 2008b | 10 | + | varied with plant type | 2000-02 |
| 281 | 50 | 50 | C. AMER. | BCI | Chave et al. 2003 | 1 | + | varied with plant type | 1985-00 |
| 307 | " | " | " | " | Chave et al., 2008a | 1 | - | Chave et al., 2005 | 1985-05 |
| 161 | 9 | 500 | C. AMER. | LS | Clark and Clark, 2000 | 10 | - | Brown, 1997 (trop. wet) | 1997 |
| 321 | 72 | 6400 | AMAZON | DUC | de Castilho et al., 2010 | 1 | - | Higuchi et al., 1998 | 2000-03 |
| 324 | " | " | " | " | de Castilho et al., 2010 | 1 | - | Higuchi et al., 1998 | 2003-05 |
| 380 | 20 | 100000 | AMAZON | BDFFP | Pyle et al., 2008 | 10 | - | Chave et al., 2005 | 1997-04 |
| 334 | " | " | " | " | Pyle et al., 2008 | 10 | - | Chambers et al., 2001 | 1997-04 |
| 281 | 20 | > 20 | AMAZON | TAP-KM67 | Vieira et al., 2004 | 35 | - | Chambers et al., 2001 | 1999 |
| 298 | " | " | " | " | Pyle et al., 2008 | 35 | - | Chambers et al., 2001 | 1999-05 |
| 394 | " | " | " | " | Pyle et al., 2008 | 35 | - | Chave et al., 2005 | 1999-05 |
| 272 | 25 | 25 | AMAZON | YASUNI | Valencia et al., 2009 | 10 | - | Chave et al., 2005 | 1995-99 |
| 282 | " | " | " | " | Chave et al., 2008a | 1 | - | Chave et al., 2005 | 1995-00 |
| 274 | " | " | " | " | Valencia et al., 2009 | 10 | - | Chave et al., 2005 | 2002-03 |
| 190 | 10 | 10 | AMAZON | RIO-BR | Vieira et al., 2004 | 35 | - | Chambers et al., 2001 | 1999 |
| 497 | 52 | 52 | ASIA | LAMBIR | Chave et al., 2008a | 1 | - | Chave et al., 2005 | 1992-03 |
| 358 | 25 | 25 | ASIA | SINHA | Chave et al., 2008a | 1 | - | Chave et al., 2005 | 1993-98 |
| 340 | 50 | 50 | ASIA | PASOH | Chave et al., 2008a | 1 | - | Chave et al., 2005 | 1986-00 |
| 290 | 25 | 25 | ASIA | PALANAN | Chave et al., 2008a | 1 | - | Chave et al., 2005 | 1999-03 |

***Coarse woody debris (CWD).*** Estimates of tropical-forest CWD span a wide range and are methods-dependent (see Table 4). The different methods in current use can produce significantly different estimates for the same site and time (e.g.,

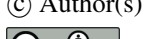



the two 2005 estimates for **JH-CLAY**, Table 4). The spatial heterogeneity of standing and fallen CWD within tropical forests calls for landscape-scale sampling. CWD stocks are also likely to significantly change through time due to the temporal variation in tree mortality in this biome (see below).

**Table 4.** Landscape-scale estimates of coarse woody debris in lowland old-growth tropical forests.
Standing dead: + indicates it was included in the CWD estimate.  When CWD was reported as Mg C, biomass is assumed 50% C.

| CWD (Mg ha$^{-1}$) | Standing dead | Measured area (ha) | Total study area (ha) | Region | Site code | Min. diam. (cm) | Method used | Year(s) | Citation |
|---|---|---|---|---|---|---|---|---|---|
| 32 | + | 20¶ | 100000 | C. AMER. | BDFFP | 10 | inventory + line-intercept | 1997-9 | Pyle et al. 2008 |
| 96 | + | 20** | 20 | AMAZON | TAP-KM67 | 2 | inventory + line-intercept | 2001 | Rice et al. 2004 |
| 50 | + | 12*** | 400 | AMAZON | JURU | 10 | inventory + line-intercept | 2003-4 | Palace et al. 2007 |
| 46 | - | ca. 0.06* | 12 | AMAZON | JH-SAND | 10 | line intercept (610 m) | 2005 | Chao et al. 2008 |
| 41 | + | 0.5 | 0.5 | " | " | " | stand-level inventory | " | " |
| 31 | - | ca. 0.06* | 12 | AMAZON | JH-CLAY | 10 | line intercept (640 m) | 2005 | Chao et al. 2008 |
| 20 | + | 1 | 1 | " | " | " | stand-level inventory | " | " |
| 53 | + | 9 | 500 | C. AMER. | LS | 10 | stand-level inventory | 1997 | Clark et al. 2002 |

¶20 ha inventoried for standing-dead stems; line-intercept used in subplots totalling 0.8 ha for fallen pieces > 10 cm dia.

\* measured area estimated as 1m x total length of transects

\*\* 20 ha for standing-dead stems; subplot line-intercepts (3.8 ha) for fallen pieces > 30 cm dia.; smaller areas for smaller pieces.

\*\*\* 12 ha inventory for standing-dead stems; line-intercept (12-km transect) for fallen pieces > 10 cm dia., smaller areas for smaller pieces

*Fine roots.* Highly-replicated, landscape-scale field observations of this C stock are potentially useful as a lower bound. Fine-root biomass is notoriously heterogeneous at multiple spatial scales. Studies within diverse tropical forests have demonstrated within-forest decreases in fine-root biomass with increasing microsite-scale availability of nutrients or water, as occurs along catenas or among the intercalated soil types in these forests (Palmiotto et al., 2004; Powers et al., 2005; Epron et al., 2006; Espeleta & Clark, 2007; Kochsiek et al., 2013; Noguchi et al., 2014; Wurzburger and Wright, 2015). Also, landscape-scale fine-root stocks can vary markedly through time. For example, fine-root stocks varied by 2.5 Mg ha$^{-1}$ over a 7-yr period in a Costa Rican wet forest (**LS** in Table 5; Espeleta & Clark, 2007). Dynamic ecosystem models would ideally hope to capture such time-series.

As illustrated in Table 5, the methods used to quantify "fine roots" vary in multiple ways, including the maximum diameter of evaluated roots, the depth of soil cores, and whether or not dead roots are included. These methods variations make cross-site comparisons and model benchmarking difficult.

A separate critical issue affects observations of fine-root stocks in all forest types, boreal to tropical: fine-root sampling in forests is usually restricted to the surface soils. No study has quantified fine roots all the way down the soil column in any tropical forest (see Table 5). The soils underlying these forests are often many meters deep.  Nepstad et al. (1994) found live roots down to at least ca. 18 m depth under one Brazilian tropical forest (**TAP-DROU** in Table 5); over the depth interval 2-6 m, fine-root density was relatively constant but much reduced compared to that of surface fine-roots. Given the great soil volume at depth, the contribution of deep fine roots both to total fine-root stocks and for ecosystem



**Table 5.** Estimates of fine-root stocks based on multiple hectares within each lowland old-growth tropical forest. Dead roots: + indicates that dead roots are included. When mass was reported as Mg C, C content is assumed to be 50%.

| Fine roots (Mg ha$^{-1}$) | Total core area, m$^2$ | Total study area, ha | Region | Site code | Max. dia. (mm) | Soil depth (cm) | Dead roots | N, cores | Year(s) | Citation |
|---|---|---|---|---|---|---|---|---|---|---|
| 5.9 | ? | ? | CARIBB. | BISLEY | 20 | 0-10 | - | ? | 2007 | Cusack et al. 2011 |
| 0.5 | 0.4 | >10 | AMAZON | TAP-SIL (clay) | 2 | 0-10 | - | 144 | 7/99-5/01 | Silver et al. 2005 |
| 0.5 | 0.4 | >10 | AMAZON | TAP-SIL (sand) | 2 | 0-10 | - | 144 | 7/99-5/01 | Silver et al. 2005 |
| 2.5*, 3.5* | ? | 2 | AMAZON | TAP-DROU | 2 | 0-10 | + | 20, 20 | 1998-9 | Nepstad et al. 2002 |
| 3.4*, 4.2* | " | " | " | " | 2 | 0-600 | + | 20, 20 | " | " |
| 12.9¶ | 0.36 | ca. 30 | AMAZON | MAN-NOG | ? (>2) | 0-40 | + | 9 | ? (pre-2014) | Noguchi et al. 2014 |
| 2.4 | 0.03 | >10 | C. AMER. | LS | 2 | 0-40 | + | 15 | 9-10/01 | Powers et al. 2005 |
| 1.1** | 1.59 | 500 | " | LS (YO) | 2 | 0-50 | - | 900** | 10/97-4/04 | Espeleta & Clark 2007 |
| 1.6** | 1.59 | 500 | " | LS (OO) | 2 | 0-50 | - | 900** | 10/97-4/04 | Espeleta & Clark 2007 |
| 5.0 | 0.03 | >10 | AMAZON | CC | 2 | 0-40 | + | 15 | 10/01 | Powers et al. 2005 |
| 2.8 | 0.03 | >10 | C. AMER. | BCI | 2 | 0-40 | + | 15 | 9-10/01 | Powers et al. 2005 |
| 8.0 | 0.03 | >10 | AMAZON | KM41 | 2 | 0-40 | + | 15 | 11/01 | Powers et al. 2005 |
| 5.6 | 0.07 | 4 | ASIA | MAEKL | 3 | 0-30 | - | 3 | 11/98 | Takahashi et al. 2012 |
| 4.5 | 0.06 | 52 | ASIA | LAMBIR | 2 | 0-10 | - | 88 | ? (pre-2013) | Kochsiek et al. 2013 |

* 2 1-ha plots, 20 cores in each, to 6 m depth

¶dead roots= ca. 13% of fine root mass; fine-root mass, Mg ha$^{-1}$ (3 cores ea.): 8.7 (plateau), 10.5 (mid-slope), 19.8 (bottom)

** 6 cores ea. in 6 0.5-ha plots on younger oxisol (YO) terraces and 6 0.5-ha plots on older oxisol (OO) plateaus; 25 dates

function may be significant in tropical forests. Models increasingly predict root stocks at different levels in the soil based on an assumed exponential decay down the vertical profile. In such cases model-data comparisons should be made for the actual soil layer of the measurements. Because all models require total root mass, however, extrapolation will be required in one domain or the other.

*Coarse roots.* There are as yet no stand-level observations of coarse roots in any forest type. In tropical forests, the field sampling for these spatially-variable organs has been confined to harvesting the root systems of selected individual trees (e.g., Niiyama et al., 2010) or to sampling coarse roots in pits or trenches away from trees, thus missing their tap roots and other large roots (e.g., Castellanos et al., 1991; Veldkamp et al., 2003). A recent survey of the available harvest data (Waring & Powers, 2017) found that root:shoot ratios for individual trees from old-growth tropical forests averaged ca. 0.65, indicating the importance of this biomass component. Notably, this ratio strongly contrasts with the 0.21 multiplier commonly used to extrapolate tropical-forest coarse-root biomass from estimated aboveground live biomass (e.g., Malhi et al., 2009; Girardin et al., 2010; Quinto-Mosquera and Moreno, 2017).

*Soil organic carbon (SOC).* SOC is strongly underestimated in all forest types (boreal to tropical) because it is rarely if ever quantified to depth (Jobbagy and Jackson, 2000). The limited tropical data in hand for subsurface SOC indicate that total SOC can dominate the C inventory in lowland tropical forests, where soils are commonly several to many meters





deep (Sombroek et al., 2000). In two tropical forests where SOC was quantified to at least 3-4 m depth (Table 6), the cumulative SOC stock to the maximum sampled depth was roughly ten times that at the surface (0-10 cm). Notably, cumulative SOC also exceeded the estimated C in aboveground live biomass (Table 6). Only in one of these cases (**LS-younger oxisol**) was SOC quantified down to the parent material. In the other two, the sampling ended many meters shy of

the total soil depth, thus missing large amounts of SOC. At the Amazonian site **PARAGOM**, where Trumbore et al. (1995) sampled SOC down to 8 m (Table 6), the soil shafts of Nepstad et al. (1994) actually extended down to 18 m depth.

The incompletely-quantified SOC is a particularly critical data gap for tropical forests. There is accumulating evidence that the huge C stocks in the deep soils underlying many of these forests are not inert (e.g., Trumbore et al., 1995, Veldkamp et al. 2003). At the Costa Rican **LS** site (Table 6), the SOC at 2-3 m depth was found to be strongly temperature-

responsive (Schwendenmann et al., 2006), indicating a vulnerability of this large tropical-forest C stock to future warming. Deep SOC (1-4 m depth) at this forest site was also found to mobilize with forest-to-pasture conversion (e.g., 30 Mg C ha$^{-1}$ lost from this subsurface soil layer in ca. 30 yr; Veldkamp et al., 2003). Changes in tropical-forest SOC, particularly in the deeper soil layers, could strongly impact the total forest C stocks and net C balance of this biome.

A second issue in tropical forests is that SOC shows marked spatial variation at all scales: from one square meter to

the next (Powers, 2006) and across the major edaphic changes (topography, soil types; see Richter and Babbar, 1991) within a forest. An example of this within-forest heterogeneity is the significant difference in cumulative SOC content between two major soil types at the **LS** site (Table 6). Distributed and replicated sampling is therefore required to quantify this important C stock.

**Table 6.** SOC estimates based on sampling to > 1 m depth in multiple ha in old-growth tropical forests. For each site, estimates are for cumulative SOC over depth range. EAB: estimated aboveground biomass.

| Cumulative SOC Mg C ha$^{-1}$ | EAB Mg C ha$^{-1}$ | Total study area (ha) | Region | Site code | Soil depth (cm) | N, cores | Year | Citation |
|---|---|---|---|---|---|---|---|---|
| 26 | 180[1] | > 10 | AMAZON | PARAGOM | 0-10 | 24 | 1992 | Trumbore et al., 1995 |
| 102 | 180[1] | " | " | " | 0-100 | 3 | " | " |
| 168 | 180[1] | " | " | " | 0-300 | " | " | " |
| 206 | 180[1] | " | " | " | 0-500 | " | " | " |
| 257 | 180[1] | " | " | " | 0-800 | " | " | " |
| 29 | 83[2] | > 50 | C. AMER. | LS-younger oxisol | 0-10 | 3 | 1999 | Veldkamp et al., 2003 |
| 123 | 83[2] | " | " | " | 0-100 | " | " | " |
| 213 | 83[2] | " | " | " | 0-300 | " | " | " |
| 35 | 74[2] | > 100 | C. AMER. | LS-older oxisol | 0-10 | 3 | 1999 | Veldkamp et al., 2003 |
| 201 | 74[2] | " | " | " | 0-100 | " | " | " |
| 330 | 74[2] | " | " | " | 0-300 | " | " | " |
| 373 | 74[2] | " | " | " | 0-400 | " | " | " |

[1] from Nepstad et al. 1994

[2] from Clark and Clark 2000



### 4.3 Ecosystem C fluxes

***Net Ecosystem CO$_2$ Exchange (NEE).***

"The eddy flux method has been criticized for uncertainty in its nighttime measurements. This is especially obvious in tropical areas, where nighttime turbulence is not well developed.   Nevertheless,... Convincing

results can be obtained from daytime eddy flux measurements..."  (Tan et al., 2013)

"It is clear that the choice whether or not to filter and replace nighttime [Amazon forest eddy-flux] data represents the single major uncertainty in the whole estimation process. The choice can turn a very large carbon sink into a moderate one or even into a small source." (Araújo et al., 2002)

When taken at short time-steps during the daytime, above-canopy measurements of the net ecosystem exchange of CO$_2$ (NEE) based on the eddy-flux (also "eddy-covariance") technique have provided valuable indications of the environmental responses of tropical-forest physiology (e.g., depression of daytime NEE at high temperatures and/or high VPD - Doughty and Goulden, 2008; Vourlitis et al., 2011). No other technique provides direct field observations of the short-term climatic

responses of forest-level CO$_2$ exchange.  Further, when daytime eddy-flux data from multiple years are filtered in a standard way (e.g., for periods of high light for estimating optimum uptake, as by Tan et al. [2013]), they can indicate how or whether these environmental responses have varied through time.

For NEE at longer time-steps (days to years), however, estimates based on the eddy-flux technique in tropical forests do not provide reference-level field benchmarks for the models. Multiple issues for this technique in these forests

create large uncertainties about the magnitude and even the sign of such estimates. The prevalence of still-air conditions at night (e.g., 70-80% of 30-min nighttime periods; Loescher et al., 2003 [Costa Rica]; Miller et al., 2004 [Brazilian Amazon]) means that the technique is inoperative or likely to be strongly biased during most nighttime periods. Studies have shown that the terrain irregularities typical of tropical forests can produce artifacts due to CO$_2$ movement into or out of an eddy-flux site through lateral advection in these still-air periods (Goulden et al., 2006; de Araújo et al., 2008; Tóta et al., 2008). In

multiple studies (Araújo et al., 2002; Saleska et al., 2003; Miller et al., 2004) the eddy-flux estimate of yearly NEE from a given year's worth of data switched from C source to C sink with different data-filtering for these periods of slow air movement. Further uncertainty in eddy-flux estimates of tropical-forest annual NEE is caused by the substantial data gaps due to heavy rainfalls, to frequent problems with instruments and with power, and due to equipment damage from animals, tree-falls, and lightning. For one forest eddy-flux study in Borneo, the actual NEE data after data-filtering covered only 30%

of the 17-mo study period (Katayama et al., 2013). Diverse methods are then used to fill the many periods of missing data (e.g., predicting daytime NEE based on radiation data [Katayama et al., 2013] or assuming a constant value for nighttime NEE [Loescher et al., 2003]).



***Gross Primary Productivity (GPP).***

"... there is no way of directly measuring the photosynthesis or daytime respiration of a whole ecosystem of interacting organisms; instead, these fluxes are generally inferred from measurements of net ecosystem-atmosphere $CO_2$ exchange (NEE), in a way that is based on assumed ecosystem-scale responses to the environment....Our [$^{13}C/^{12}C$] analysis indicates that daytime ecosystem respiration differed fundamentally from standard predictions that were based on nighttime NEE and temperature... " (Wehr et al., 2016)

As underlined in the above quote, no method exists for directly observing total forest-level photosynthesis (also termed "Gross Primary Productivity" or GPP). The existing field estimates of tropical-forest GPP have been derived based on modeling, assumed physiology, extrapolation and/or incomplete field observations. Benchmark-level direct field observations are therefore lacking for this critically-important C flux.

Although GPP estimates have been produced by tropical-forest eddy-covariance studies, the sole $CO_2$ flux that is actually assessed with that technique is NEE, the small difference between two much larger, opposing fluxes (GPP and ecosystem respiration, $R_{eco}$). As discussed above, eddy-flux NEE data from tropical-forests are themselves highly uncertain and incomplete. The standard current approach for "partitioning" NEE into GPP and $R_{eco}$ is based on assumptions about forest ecophysiology that have recently been challenged by findings from parallel $^{13}C/^{12}C$ measurements in a temperate forest (Wehr et al., 2016).

Alternatively, bottom-up biometric approaches have been used to estimate GPP for some tropical-forest sites (e.g., Doughty et al., 2013; Malhi et al., 2015). These studies, carried out in a single 1-ha plot per forest, have been based on combining sparse direct observations of some components of production and respiration with intuitive estimates for, or omission of, many unmeasured components (see section 2.1. and Table 7). In tropical forests, the summed C in the unmeasured processes may equal a significant fraction of total GPP (Clark et al., 2001a; Litton and Giardina, 2008).

***Ecosystem respiration ($R_{eco}$).*** Similarly, existing eddy-flux estimates for whole-forest respiration in this biome remain questionable due to multiple issues: 1) the uncertainty of the NEE estimate from which $R_{eco}$ is inferred (see above); 2) the likelihood of lost (/extra) respiration due to lateral advection of $CO_2$ during the predominantly still nights (Goulden et al., 2006; Tóta et al., 2008); and 3) unresolved questions about the assumptions underlying the estimation of daytime $R_{eco}$ from NEE (Chambers et al., 2004; Wehr et al., 2016; Wohlfart et al., 2017).

***Autotrophic respiration ($R_a$) and heterotrophic respiration ($R_h$).*** Benchmark-level field observations of these two fractions of $R_{eco}$ are as yet lacking for tropical forests. Neither of these fluxes can be directly field-assessed at the ecosystem level. Some estimates of stand-level $R_a$ (e.g., Doughty et al., 2015 and included references) have been derived for different tropical forests in the Global Ecosystem Monitoring (GEM) project. These estimates were based on sparse field measurements in a single hectare of the studied forest, of a subset of $R_a$ components (fine-root respiration [estimated as soil $CO_2$ efflux minus that with root exclusion], canopy-leaf dark respiration, and tree-bole $CO_2$ efflux). These measurements





were then combined with intuitive estimates for two unmeasured $R_a$ components (daytime leaf respiration, respiration by coarse roots). The substantial $CO_2$ efflux from small-diameter wood (< 10 cm diameter) was not considered; however, in a Costa Rican forest this $R_a$ component was estimated to account for 70% of total woody $CO_2$ efflux, based on extensive sampling from mobile climb-up towers (Cavaleri et al., 2006). In the soil, the intimate inter-relations among roots, root exudates, root symbionts, and soil microbes make the distinction between $R_h$ and $R_a$ both conceptually and methodologically challenging (Trumbore, 2006). An aspect of $R_h$ that is rarely measured in tropical forests is the $CO_2$ efflux from decomposing coarse woody debris. This respiration component has been estimated at 6-16% of total tropical-forest $R_{eco}$, based either on extrapolating spot field measurements of respiration from CWD to the stand level (Chambers et al., 2004 [Central Brazilian Amazon]) or on combining landscape-scale estimates of CWD stocks with inferred CWD turnover-time (Hutyra et al., 2008 [Eastern Brazilian Amazon], Cavaleri et al., 2008 [Costa Rica]).

   *Total net primary productivity (Total NPP).* No benchmark field observations are available for Total NPP. As is the case in all other forest types (Clark et al., 2001a), the field studies in tropical forests have been restricted to a subset of NPP components (Table 7). Those that remain unquantified could sum to a substantial fraction of Total NPP (see also Clark et al., 2001a,b; Litton and Giardina, 2008; Cleveland et al., 2015). For the models, the sum of the field-assessed NPP components provides a lower bound for Total NPP.

   Two NPP constituents so far missing from the field studies (Litton and Giardina, 2008) and from most models (Fatichi et al., 2014) are the amounts of new fixed C being lost (exported) from the plants belowground, either to root symbionts (nodules and/or mycorrhizae) or to the soil through root exudation. Isotopic evidence from a $CO_2$ enrichment study in a temperate forest indicated the likelihood of significant C export from the roots; they found belowground transfer of a substantial fraction of the assimilated C, with strong signals in mycorrhizal sporocarps and in soil respiration (a mix of $R_h$ and $R_a$) but not in the fine roots (Steinmann et al., 2004). Because most tropical trees support mycorrhizae (Janos, 1980), and legumes, potential N-fixers, are present in most tropical forests, the possibility exists of considerable allocation of NPP to symbionts. This aspect of C cycling is practically unstudied in the biome. In one exceptional study in a Costa Rican forest (Lovelock et al., 2004), extraradical hyphal production by arbuscular mycorrhizae at 0-10 cm soil depth was estimated at 1.5-1.9 Mg ha$^{-1}$ yr$^{-1}$. Because the total plant-assimilated C going into new mycorrhizal fungal tissues also includes that incorporated into spores and sporocarps, the hyphae inside roots, and all the hyphae in the soil below 10-cm depth, this NPP component appears to be significant in this forest. Root exudation, as yet unstudied, is another potentially non-trivial portion of tropical-forest NPP.

   Opportunities for data-model fusion will be maximized by developing the C-cycle models to explicitly specify those NPP components that have been field-assessed. As recently reported by Negrón-Juárez et al. (2015), only three of the ten



**Table 7.** The biometric components of Total NPP in tropical forests (Mg C ha$^{-1}$ yr$^{-1}$). Observed ranges (bold) are from examples in this paper and in Clark et al. 2001b. Guesstimates (italics) are for components as yet unquantified in tropical forests.

| Component | Observed range | Guesstimate | Comment |
|---|---|---|---|
| VOC (volatile organics) production | | *0.1 - > 0.9* | Likely increase in Isoprene prod. with warming |
| Aboveground wood production (larger stems) | **1.0 - 3.8** | | Unverified estimates via off-site allometries |
| Wood prod. by smaller stems + hemiepiphytes | | *≤ 0.1 - 0.38* | Rarely if ever quantified |
| Branch-shedding by live trees | | *0.1 - 3.0* | Requires distinguishing pieces from dead trees |
| Twig litterfall (twigs ≤ 1 cm in diam.) | **0.4 - 1.3** | | Likely underestimate (pre-collection decomp.) |
| Leaf litterfall | **2.9 - 3.4** | | The surrogate for actual leaf production |
| Leaf mass lost to herbivory | | *0.6 - 1.1* | Increasing with rising [$CO_2$] and C:N, C:P? |
| Leaf mass lost to decomposition, leaching | | *0.1 - 1.0* | Signif. pre-collection losses in tropical forests |
| Reproductive litterfall | **0.2-0.7** | | |
| Reproductive losses to consumers | | *≥ 0.1 - 0.8* | Fruits are animal-dispersed, made to be eaten |
| Reproduction lost to pre-collection decomposition | | *0.1 - 0.3* | |
| New non-structural CHO's (stores) | | *?* | |
| Coarse-root production | | *0.2-2.3* | |
| Surface-soil fine-root production (0-30 cm) | **0.3 - 0.9** | | |
| Deeper fine-root production (0.3m to depth) | | *0.1-0.5* | |
| Fine-root losses to herbivory & decomp. | | *>> 0* | As yet unstudied; possibly non-trivial |
| C exports to root symbionts (mycorrhizae, nodules) | | *>> 0* | A signif. NPP fraction in most tropical forests? |
| Root exudates | | *>> 0* | A large NPP fraction? Rising with [$CO_2$]? |

ESMs in the Coupled Model Intercomparison Product (CMIP5) report "leaf NPP", "wood NPP" and "root NPP". The

15 different production components are functionally distinct. In a landscape-scale field study at the Costa Rican LS site, the several field-quantified NPP components varied independently through 12 years, showing distinct relationships to the interannual variation in temperature, rainfall, and VPD (Clark et al., 2013). Below, we consider individually those biometric NPP components that have been assessed to date in tropical lowland forests.

*Fine litterfall* In tropical forests, biometric aboveground NPP is typically dominated by short-lived tissues (Clark

20 et al., 2001a). These are assayed as shed "fine litterfall" collected in litter traps (Table 8). Fine litterfall varies spatially

**Table 8.** Landscape-scale estimates of the components of fine litterfall (leaf, reproductive, twig) in lowland old-growth tropical forests. Grd. traps: +/- indicates whether ground-level traps were used to collect large items (e.g., 3-m palm leaves); if not, leaf litterfall is likely to be underestimated.

| Fine litterfall (Mg ha$^{-1}$ yr$^{-1}$) | | | Twig diam. (cm) | Trap area (m$^2$) | Study area (ha) | Grd. traps | Region | Site code | Citation | Years |
|---|---|---|---|---|---|---|---|---|---|---|
| Leaf | Reprod. | Twig | | | | | | | | |
| 5.7 | 0.7 | 1.4 | ? | 60 | 50 | - | GUIANAS | PISTE-ST.E | Puig and Delobelle, 1988 | 1978-1981 |
| 5.8 | 0.7 | 1.8 | <1 | 30 | 10 | - | GUIANAS | NOU-PP | Chave et al., 2008b | 2001-2007 |
| 6.6 | 0.8 | 2.5 | <1 | 50 | 12 | - | GUIANAS | NOU-GP | Chave et al., 2008b | 2001-2007 |
| 6.8 | 1.3 | 0.9 | <1 | 81 | 500 | + | C. AMER. | LS | Clark et al., 2013 | 1997-2009 |
| 6.4 | 0.6 | 1.4 | ? | 17 | ca. 10 | - | C. AMER. | BCI | Leigh et al, 1990 | 1972-1979 |





within each tropical forest. When assessed in 18 0.5-ha plots distributed within one neotropical forest (**LS,** Table 8), the plots differed (max - min) in annual leaf litterfall by 3.8 to 6.3 Mg ha$^{-1}$ yr$^{-1}$, depending on the year; for reproductive litterfall, the across-plot range was > 2 Mg ha$^{-1}$ yr$^{-1}$ in most of the 12 years (data in Table S2 in Clark et al., 2013). Landscape-scale data are therefore needed for reference-level benchmarks for this aspect of tropical-forest C cycling. Because the three

components of fine litterfall are functionally distinct, they are considered individually below.

       *Leaf litterfall (vs. leaf production).* In field studies of biometric NPP (termed NPP\*, Clark et al. 2001a), leaf litterfall over a given study interval is typically taken as a surrogate for leaf production over that interval. Stand-level leaf production itself has not been quantified in the field in tropical forests. In most tropical forests, leaf litterfall is the largest contributor to aboveground NPP\* (Clark et al., 2013 and included references). It can be a misleading surrogate for leaf

production in terms of both mass and timing. One methods issue is the difficulty of quantifying the very large fallen leaves in tropical forests (e.g., 3-m long palm leaves). Ground-level and/or very large traps are required to collect these large items of "fine litter" (Villela & Proctor, 1999) but are rarely used. In addition, in tropical forests leaf litterfall undervalues leaf production due to two types of pre-collection losses (Table 7; also see Clark et al., 2001b). One is the mass loss from pre-collection decomposition and leaching of the shed leaves in the hot, humid conditions. Some leaves hang up in the

vegetation and decompose above the ground. When Frangi and Lugo (1985) suspended old leaves from palms in a Puerto Rican forest, they found that roughly half the leaf mass was lost through decomposition in four months. A second issue is the leaf mass removed by herbivores (Table 7). Partial leaf damage (holes in fallen leaves) was estimated at ca. 0.8 Mg C ha$^{-1}$yr$^{-1}$ in a lowland Peruvian forest (Metcalfe et al., 2013); in addition, leaf-monitoring studies (Lowman et al., 1984, Filip et al., 1995) have shown that an equivalent amount or more may typically be lost to herbivores that remove entire leaves.

20       One potential approach for models would be to explicitly include the processes of herbivory and decomposition losses that occur between leaf production and leaf shedding, therefore facilitating a direct comparison. In lieu of this, model-data comparisons should take into account the low bias of leaf-litterfall observations. In cases where leaf litterfall is conflated with leaf production for the purposes of determining allocation to the leaf fraction, the resulting allocation underestimate might lead to underestimating LAI.

25       A separate issue is that the seasonal timing of leaf production can differ from that of leaf litterfall, as found by Reich et al. (2004) in a Venezuelan tropical forest (in most species studied, although there was some degree of correlation). In many tropical forests, leaf litterfall typically peaks at the time of the yearly maximum soil dry-down (Wagner et al., 2016); this timing can be distinct from that of actual leaf production. Such a timing disjunct will complicate attempts to evaluate the seasonality of tropical-forest NPP and C allocation when leaf litterfall is used as the surrogate for production

(e.g., Doughty et al., 2013).

       *Twig litterfall (vs. twig production).* Estimates of twig litterfall should be treated as a lower bound for twig production. In tropical forests, twig litterfall (Table 8) is likely to strongly underestimate actual production due to substantial mass loss before collection. In a New Guinea rain forest, when Edwards (1977) compared canopy-collected live twigs < 1





cm dia. to <1 cm dia. twigs in the litter traps, the fallen twigs were found to have already lost 36-40% of their mass, presumably due to decomposition and/or leaching when they were still attached to the branches above.

   ***Reproductive litterfall (vs. reproductive production).*** The biometric surrogate for reproductive production, reproductive litterfall (Table 8), is likely to undervalue production by at least 100%. This NPP component is not easily

quantified at the stand level. Tropical forests are typically dominated by animal-dispersed plants. The consumers are likely to remove most of the fruits produced, leaving the "crumbs" to fall into the litter traps. In a Puerto Rican palm forest, for example, fruit production assessed by direct observation over time exceeded the fruit mass in littertraps by a factor of 14 (Lugo and Frangi, 1993). Similarly, in a Colombian tropical forest, the estimate of fruit production based on observing from platforms and from climbing ropes was double the estimate based on fruit mass in the litter traps (Parrado-Rosselli et al.,

10   2006).

   For multiple reasons, this NPP component merits attention for the models. Many Land Surface Models do not specifically include the carbon allocation to reproduction; this omission implies corresponding overestimates of stocks of other carbon pools (e.g., roots, stems, leaves). Demographic models, in contrast, typically do specify reproductive allocation, which is needed to drive forest recruitment (Moorcroft et al., 2001). Secondly, reproductive tissues are nutrient-rich (e.g., in

nitrogen, phosphorus, and cations) and thus likely play a significant role in the cycling of those nutrients. Reproductive status could influence nutrient resorption and thus re-allocation of carbon (Tully et al., 2013). A third issue is that this production component could be responding to climatic/[$CO_2$] changes. Two recent tropical-forest studies suggest multi-decadal increases in forest-level reproduction (reproductive litterfall - Clark et al., 2013; flowering incidence - Pau et al., 2013).

***Aboveground wood production (EABI)***   As for aboveground woody biomass (above), field estimates of aboveground wood production, also termed EABI (Estimated Aboveground Biomass Increment), are unverified and highly uncertain. This production component is based on measurement at two successive censuses of the diameters of all live stems in the study plot that exceed an arbitrary diameter limit (usually 10 cm); these data are then used for allometric estimation of the tree's aboveground biomass at both times. EABI is calculated as the sum of the estimated biomass increments by all the

stems that survived the interval, plus the estimated increments above the specified size limit by the recruits, those smaller stems that grew past the minimum size by the second census (see Clark et al., 2001a). One methods variant (Chave et al., 2008b; Pyle et al., 2008), equating the census-interval growth by new recruits to their total estimated mass at the second census, substantially overestimates these small trees' contribution to stand growth; before reaching the 10-cm diameter limit, most small trees in tropical forests have grown very slowly over decades (see Clark and Clark, 2001; Rozendaal et al., 2015).

As for estimates of aboveground biomass, because EABI depends on an unverified allometric relationship between




**Table 9.** Landscape-scale estimates of aboveground wood production (EABI, Mg ha$^{-1}$ yr$^{-1}$) in lowland old-growth tropical forests. Int. length: the length of the interval between censuses.  Min. dia.: the minimum diameter of the measured stems in each census

| EABI | Plot area (ha) | Study area (ha) | Region | Site code | Citation | Min. dia. (cm) | Allometry used | Method for recruit growth | Int. length (yr) | Years |
|---|---|---|---|---|---|---|---|---|---|---|
| 8.3[1] | 20 | ? | AMAZON | TAP-KM67 | Pyle et al., 2008 | 10 | Chave et al., 2005 | est. biomass[3] | 2-4 | 1999-05 |
| 7.2[1] | " | " | " | " | " | " | Chambers et al., 2001 | " | " | " |
| 6.6 | 20 | 100000 | AMAZON | BDFFP | Pyle et al., 2008 | 10 | Chave et al., 2005 | est. biomass[3] | 5 | 1997 - 04 |
| 5.7 | " | " | " | " | " | " | Chambers et al., 2001 | " | " | " |
| 8.7 | 12 | 12 | GUIANAS | NOU-GP | Chave et al., 2008b | 10 | Chave et al., 2005[2] | est. biomass[3] | 8 | 1992-02 |
| 8.0 | 10 | 10 | GUIANAS | NOU-PP | Chave et al., 2008b | 10 | Chave et al., 2005[2] | est. biomass[3] | 8 | 1992-02 |
| 3.7 | 9 | 500 | C. AMER. | LS | Clark et al., 2013 | 10 | Brown et al., 1997 | inc. > 10 cm[4] | 1 | 1997-98 |
| 5.0 | " | " | " | " | " | " | " | " | " | 2005-06 |
| 5.0 | 50 | 50 | C. AMER. | BCI | Chave et al., 2008a | 1 | Chave et al., 2005 | ? | 5 | 1985-05 |
| 6.8 | 24 | 24 | AMAZON | YASUNI | Chave et al., 2008a | 1 | Chave et al., 2005 | ? | 5 | 1995-00 |
| 7.0 | 50 | 50 | ASIA | PASOH | Chave et al., 2008a | 1 | Chave et al., 2005 | ? | 5 | 1986-00 |
| 7.2 | 52 | 52 | ASIA | LAMBIR | Chave et al., 2008a | 1 | Chave et al., 2005 | ? | 5 | 1992-03 |
| 4.9 | 16 | 16 | ASIA | PALANAN | Chave et al., 2008a | 1 | Chave et al., 2005 | ? | 4 | 1999-03 |
| 7.4 | 25 | 25 | ASIA | SINJA | Chave et al., 2008a | 1 | Chave et al., 2005 | ? | 5 | 1993-98 |

[1] stems 10-<35 cm diameter measured in subplots totalling 4 ha; stems ≥ 35 cm diameter measured over 20 ha

[2] for trees; for lianas, used allometry of Schnitzer et al., 2006

[3] the contribution to EABI from recruits is defined as their total estimated biomass

[4] the contribution to EABI from recruits is defined as their estimated growth above 10-cm diameter

stem diameter and stem biomass, all values of this metric involve unquantifiable uncertainty. When different allometries are applied to the same set of diameter data, different estimates of EABI can be produced (e.g., duplicate estimates at site **TAP-KM67**, Table 9). Determining which if any of such estimates is reasonable would require follow-up on-site verification of the underlying allometry (Clark and Kellner, 2012).

Given the heterogeneity of biomass dynamics within a tropical forest, data-model fusion exercises and site-level model testing call for landscape-scale field data for EABI. The exception to this are those demographic models (e.g., ED, Moorcroft et al., 2001) that explicitly address the effects of the small-scale spatial heterogeneity within a forest landscape.

In spite of this metric's unquantifiable uncertainty, when estimated at the landscape scale and in the same way over a long series of successive periods, repeated annual estimates can provide valuable guidance for the models with respect to both long-term trends in this productivity component and its climatic/[CO$_2$] responses. For example, 12-yr records of EABI from the **LS** site revealed highly-significant sensitivities of landscape-scale EABI to the inter-year changes in nighttime temperatures, VPD and [CO$_2$] (Clark et al., 2013).

*Fine-root production.* The field estimates of fine-root production in tropical forests can be used as a rough lower bound. Due to the methods challenges, fine-root production has not been well-quantified in any forest type, boreal to tropical. In the tropical-forest biome, because of the notorious variation in fine-root stocks at all spatial scales (Espeleta and





Clark, 2007; Powers et al., 2005), robust assessment of fine-root production for a given forest would require highly-replicated and distributed sampling. Unfortunately, this production component has only rarely been assessed in multiple hectares of a tropical forest (Table 10). A second critical limitation is that the field measurements to date in this biome have been confined to the surface soil (0 to ≤ 30 cm depth). There are no field observations from tropical forests of production by

the deeper fine roots (live fine roots were found to at least 18 m depth in one Amazon forest; Nepstad et al., 1994).

Variable methods for assessing fine-root production (different soil depths and root sizes, inclusion or not of dead roots; Table 10) also make cross-site comparisons difficult. The usual approach in tropical forests, in-growth cores, is likely to strongly underestimate production due to lags before root in-growth and the likelihood of roots dying and decomposing before soil cores are retrieved; in a temperate pine forest, production estimates based on in-growth cores averaged 54% lower

than those from minirhizotrons (Hendricks et al., 2006). Whether root herbivory removes a significant fraction of fine root production (Lauenroth, 2000) is as yet unstudied in tropical forests.

**Table 10.** Estimates of fine-root production (Mg ha$^{-1}$yr$^{-1}$) from multiple hectares within lowland old-growth tropical forests.

| Fine-root prod. | Measured Area (m$^2$) | Study Area (ha) | Region | Site code | Root diam. (mm) | Method | Depth (cm) | Time to retrieval (mo) | Citation | Years |
|---|---|---|---|---|---|---|---|---|---|---|
| 0.7 | 0.04 | ?[1] | C. AMER. | EARTH | < 2 | in-growth cores | 0-10 | 24 | Alvarez-Clare et al., 2013 | 2008-10 |
| 3.5[2], 3.3[2] | 0.28 | 2[2] | ASIA | LAMBIR | < 10 | in-growth cores | 0-30 | 3 | Kho et al., 2013 | 2009-09 |

[1]data from the control plots of a fertilization experiment, one in each of four blocks separated by ≥ 50m

[2]data from, respectively, 1 ha in clay soil and 1 ha in sandy soil; cores extracted every 3 mo over 1 year

**4.4 Tree mortality**

"... [in a steady-state landscape] about 98.0 to 99.7% of forest land is in a carbon-sequestering stage; the

remaining 0.3 to 2% is emitting carbon...from natural breakdown (tree death, gap formation), disturbance (wind break, fire), ...pest outbreak... Unless sensors capture such short-term "emission" events ..., they will commonly signal net carbon uptake... Plot-based carbon flux measurements...cannot produce a realistic picture of a landscape's contribution to carbon sequestration. " (Körner, 2003)

"...a more comprehensive sampling scheme that includes large-area data (e.g., large plots and remote sensing) and robustly characterizes disturbance size distribution is required to understand tropical forest dynamics and its impact on carbon balance. " (Di Vittorio et al., 2014)

Biomass losses from tree mortality are a critical determinant of forest biomass stocks (McDowell et al. 2011). In tropical

forests, strong spatiotemporal variation in these losses makes quantifying and tracking them highly challenging. Illustrating





**Table 11.** Estimated mortality-driven biomass loss (Mg ha$^{-1}$ yr$^{-1}$) from multiple-ha samples in lowland old-growth tropical forests. Meas. Area: plot area where all stems were measured. Int. (yr): interval between censuses.

| Mortality biomass loss | Meas. area (ha) | Study area (ha) | Region | Site code | Citation | Minimum stem diam. (cm) | Allometry used | Int. (yr) | Years |
|---|---|---|---|---|---|---|---|---|---|
| 15*, 1* | 2* | 52 | ASIA | LAMBIR | Kho et al., 2013 | 10 | Chave et al., 2005 | 5 | 1992-97 |
| 5*, 15* | " | " | " | " | " | " | " | 6 | 1997-03 |
| 5*, 2* | " | " | " | " | " | " | " | 5 | 2003-08 |
| 6.1 | 52 | 52 | ASIA | LAMBIR | Chave et al., 2008a | 1 | Chave et al., 2005 | 5 | 1992-03 |
| 4.7 | 16 | 16 | ASIA | PALANAN | Chave et al., 2008a | 1 | Chave et al., 2005 | 4 | 1999-03 |
| 8.4 | 25 | 25 | ASIA | SINJA | Chave et al., 2008a | 1 | Chave et al., 2005 | 5 | 1993-98 |
| 5.4 | 50 | 50 | ASIA | PASOH | Chave et al., 2008a | 1 | Chave et al., 2005 | 4 | 1986-00 |
| 5.3 | 50 | 50 | C. AMER. | BCI | Chave et al., 2008a | 1 | Chave et al., 2005 | 5 | 1985-05 |
| 6.2 | 24 | 24 | AMAZON | YASUNI | Chave et al., 2008a | 1 | Chave et al., 2005 | 5 | 1995-00 |

*data from, respectively, 1 ha in clay soil & 1 ha in sandy loam soil within the 52-ha plot; from Fig. 2 in Kho et al. 2013

this variation are the contrasting losses from two 1-ha plots in a Borneo forest in each of three intervals (**LAMBIR** site, Kho et al. 2013; Table 11). Tropical-forest disturbance regimes predominantly involve frequent small-scale canopy gaps (< 150 m$^2$) caused by branch-falls or tree-falls; larger forest openings from storms, blowdowns, or extreme drought are increasingly rare in time and space as they increase in size (Chambers et al., 2013; Magnabosco Marra et al., 2014; Marvin et al., 2014; di Vittorio et al., 2014). A study in the Central Amazon combining remote sensing and ground observations (di Vittorio et al., 2014) found mortality losses to follow a power-law distribution with disturbed area, up to and including the region's extremely large blowdowns; these researchers concluded that the biomass losses observed solely in existing plots would be an inaccurate indicator (biased low) of landscape-scale dynamics. A separate complication is the disproportionate influence on biomass stocks from the deaths of scattered very large trees. In French Guianan old-growth forest (Rutishauser et al., 2010), such tree deaths were found to largely drive the heterogeneity in biomass dynamics among plots and through time. Unsurprisingly, given these sources of variation, Galbraith et al. (2013) found a six-fold variation among wood turnover rates (23-129 yr) calculated from individual small tropical-forest plots. Landscape-scale field observations are clearly required to guide the models with respect to tropical-forest mortality and its counterpart, biomass turnover. Parallel monitoring of larger forest expanses with remote sensing would further improve such estimates.

An observational finding important for the C-cycle models is the strong temporal variation in tropical-forest tree mortality. Mortality spikes have been observed in both Neotropical and Asian tropical forests in extreme climatic events such as the strong El Niño's of 1982/3 and 1997/8 and the 2005 Amazon drought (Clark, 2004; Williamson et al., 2001; van Nieuwstadt and Sheil, 2005; Phillips et al., 2009).

Some models specify stochastic dynamics of tree death (Fyllas et al., 2014; Smith et al., 2014). Many models attempt to simulate the responses of tree mortality to changes in vegetation stress (McDowell et al., 2013; Powell et al.,




2013) but more aggregated models typically use a simple turnover parameter (Galbraith et al., 2013, reviewed by McDowell et al., 2013). Introducing more robust mortality benchmarks based on combining structured ground data with satellite observations (e.g., Kellner and Hubbell, 2017) and also explicitly linking large mortality losses to extremes of climatic stressors (e.g., Phillips et al., 2009) should help modellers move towards a more process-based representation of tropical-

forest mortality.

## 4.5 Directional trends and climatic/[$CO_2$] responses of C cycling

A valuable class of benchmarks for the C-cycle models will be landscape-scale field observations of the decadal changes in and climatic/$CO_2$ responses of C stocks and fluxes in tropical forests. Given the complexities described above for quantifying forest C stocks and fluxes across time and space, detecting incremental changes caused by external drivers is a

particularly difficult problem. Long series of landscape-scale measurements at annual or greater intervals are rare for this biome.

   To illustrate this type of response benchmarks, Table 12 lists the significant relationships revealed by a 12-yr landscape-scale study of annual biometric Aboveground NPP (ANPP*) in a Costa Rican forest (Clark et al., 2013). Through that period, one of the four biometric ANPP* components, EABI, showed highly significant negative impacts from two

climatic stressors and a small positive response to increasing [$CO_2$]. One other production component, reproductive litterfall, also showed a small positive association with [$CO_2$]. Replicating such quantitative analyses across the biome and through coming decades would greatly contribute to more accurate C-cycle models for these forests. The long-term yearly C-cycle studies that have now been implemented in many large tropical-forest plots of the CTFS network (Anderson-Teixeira et al., 2015) are a major step in that direction.

**Table 12.** Climatic and [$CO_2$] responses ($\pm$ 95% confidence intervals) of C-cycling in lowland old-growth tropical forests. EABI (estimated aboveground wood production) and reproductive litterfall are in units of Mg ha$^{-1}$ yr$^{-1}$.

| Aspect, C-cycling | Response | $P$ | N, years | Site code | Years | Citation |
|---|---|---|---|---|---|---|
| EABI | - 0.95 $\pm$ 0.37 per $^{o}$C increase in year mean of daily $T_{min}$ | .00015 | 12 | LS | 1997-09 | Clark et al., 2013 |
| " | - 0.03 $\pm$ 0.01 per % incr. in hrs of VPD > 1 kPa, dry season | .00015 | " | " | " | " |
| " | + 0.021 $\pm$ 0.015 per additional ppmv of annual [$CO_2$] | .006 | " | " | " | " |
| Reproduct. litterfall | + 0.012 $\pm$ 0.011 per additional ppmv of annual [$CO_2$] | .01 | " | " | " | " |





### 4.6 Local meteorology

Sparse and intermittent climatic monitoring in all tropical regions makes the interpolated global gridded climatic datasets unreliable for this biome (see Deblauwe et al., 2016). For this reason, high-quality climatic records from tropical-forest field sites would be particularly important resources for model data-fusion exercises and thus merit inclusion among the

benchmark field observations of the ILAMB effort. For a catalog of such local climatic records, key accompanying information should include whether the data are from a ground-level met station or from above-canopy sensors, and whether the records have been screened, corrected to maintain internal consistency, and gap-filled. At the example site in Table 13, multiple adjustments to the records were required after the manual instruments were re-located and then augmented with an automated system (see Clark and Clark, 2011). The calculation ($[T_{max} + T_{min}]/2$) used in the early record to estimate daily

$T_{mean}$ from max/min thermometer data was found to significantly differ from the actual logged daily $T_{mean}$ at this site. Splicing the prior estimated record to the current record of logged $T_{mean}$ would have spuriously indicated an abrupt $1^o$ C "cooling" in the site's $T_{mean}$ record (see Fig. 2 in Clark and Clark, 2011). The long-term record for $T_{mean}$ was therefore confined to the automated data. The early records for rainfall and $T_{max/min}$ also required adjustment by cross-site and/or cross-sensor regression. Such issues likely affect many local met records from tropical-forest field sites. The longer records are

likely to include periods both before and after the introduction of an automated station. At many sites, station siting is also likely to have changed over time.

**Table 13.** Local meteorological records for lowland old-growth tropical forests (1 example site).

Qa/Qc: +, documented quality control; Cons.: +, adjusted for internal consistency over total record; Gaps: +, missing data for some periods. Location: sensors on a ground-level station (grnd) or above-canopy tower (ab-can).

| Site code | Time-step | Climatic metric | Loca-tion | Qa/Qc | Gaps | Cons | Time period | Weblink or other data source |
|---|---|---|---|---|---|---|---|---|
| LS | daily | rainfall | grnd | + | + | + | 1/1963-1992 | www.ots.ac.cr/meteoro/default.php?pestacion=2 |
| LS | daily | rainfall | grnd | + | - | + | 9/1992-2016 | " |
| LS | daily | radiation (pyr) | grnd | + | + | - | 3/1992-2016 | " |
| LS | daily | max $T_{air}$, min $T_{air}$ | grnd | + | - | + | 4/1982-2016 | " |
| LS | daily | mean $T_{air}$ | grnd | + | - | + | 3/1992-2016 | " |
| LS | 30-min | radiation (pyr, PAR) | grnd | + | + | - | 3/1992-2016 | on request to deborahanneclark@gmail.com |
| LS | hourly | $T_{air}$, RH, rainfall | grnd | + | + | - | 6/1992-2016 | " |
| LS | 30-min | $T_{air}$, RH, rainfall | grnd | + | + | - | 1/2003-2016 | " |



**Table 14.** Site codes and descriptors for the field sites in the benchmark data tables.
MAP: mean annual precipitation; MAT: mean annual temperature.

| Site Code | Region | Study site | Citation | Elevation (m) | Lat. | Long. | MAP mm | Years of MAP data | MAT °C |
|---|---|---|---|---|---|---|---|---|---|
| AGP-01,02 | AMAZON | Amacayacu, Colombia | Jiménez et al., 2014 | | 3°43'S | 70°18'W | 3342 | 1973-2008 | 26 |
| BDFFP | AMAZON | N of Manaus, Brazil | Pyle et al., 2008 | | 2° 30' S | 60° --' W | 2285 | | |
| CC | AMAZON | Cocha Cashu Stn, Peru | Powers et al,. 2005 | | 11° 54' S | 71° 72' W | 2165 | ?- pre-2004 | |
| CAX-06 | AMAZON | Caixuana, Brazil | Marthews et al., 2012 | | -1.729167 | -51.473611 | 2272 | | |
| CAX-CTL | AMAZON | Caixuana, Brazil | Metcalfe et al., 2010 | | -1.729167 | -51.473611 | 2272 | | |
| DUC | AMAZON | Reserva Ducke, Brazil | de Castilho et al., 2010 | 40-140 | 2° 55' S | 59° 59' W | ca. 2300 | ?- pre-2010 | ca. 26 |
| JH-CLAY | AMAZON | Jenaro Herrera, Peru | Chao et al., 2008 | | 4° 55' S | 73° 44' W | 2500-2700 | ?- pre-2001 | 26-27 |
| JH-SAND | AMAZON | Jenaro Herrera, Peru | Chao et al., 2008 | | 4° 55' S | 73° 44' W | 2500-2700 | ?- pre-2001 | 26-27 |
| JURU | AMAZON | Juruena, Brazil | Palace et al., 2007 | | 10°49'S | 58° 48' W | | | |
| KM41 | AMAZON | KM41 reserve, Brazil | Powers et al., 2005 | | 2° 30' S | 60° 0' W | 2650 | ?- pre-2001 | |
| MAN-NOG | AMAZON | 30 km N of Manaus, Brazil | Noguchi et al., 2014 | | 2° 36' S | 60° 8' W | | | |
| MAN-K34 | AMAZON | Manaus K34 tower, Brazil | Marthews et al., 2012 | | | | 2285[1] | 1961-1990[1] | |
| MAN-McW | AMAZON | N of Manaus, Brazil | McWilliam et al., 1993 | | | | 2285[1] | 1961-1990[1] | |
| RIO-BR | AMAZON | Rio Branco, Acre, Brazil | Vieira et al., 2004 | | 10° 07' S | 67° 62' W | 1940[2] | 1969-1990[2] | |
| PARAGOM | AMAZON | Paragominas, Para, Brazil | Trumbore et al., 1995 | | 2° 59' S | 47° 31' W | 1750 | ?- pre-1994 | |
| TAP-A1,A4 | AMAZON | Tapajos, Para, Brazil | Aragão et al., 2005 | | 2° 51' S | 54° 58' W | 1909[3] | 1967-1990[3] | |
| TAP-DROU | AMAZON | Tapajos, Brazil-drought expt. | Nepstad et al., 2002 | | 2.9° S | 54.95° W | 2000 | ?- pre-2002 | |
| TAP-KM67 | AMAZON | Tapajos, Brazil-tower site | Pyle et al., 2008 | | 2° 51' S | 54° 58' W | 1909[3] | 1967-1990[3] | 25 |
| TAP-SIL | AMAZON | Tapajos, Brazil | Silver et al., 2005 | | 2° 64' S | 54° 59' W | 1909[3] | 1967-1990[3] | 25 |
| YASUNI | AMAZON | Yasuní, Ecuador | Valencia et al., 2009 | 216-248 | 0° 41' S | 76° 24' W | 3100 | | |
| ZAR-01 | AMAZON | Zafire, Colombia | Jiménez et al., 2014 | | 4°0'S | 69°53'W | 3342 | 1973-2008 | 26 |
| BCI | C. AMER | Barro Colorado I., Panama | Chave et al., 2003 | 120-160[4] | 9° 15' N[4] | 79° 85' W[4] | 2637 | 1929-2001 | |
| LS | C. AMER | La Selva, Costa Rica | Clark, D.A. et al., 2013 | 37-150 | 10° 26' N | 83° 59' W | 4537 | 1997-2009 | 25.1 |
| EARTH | C. AMER | EARTH Univ., Costa Rica | Alvarez-Clare et al., 2013 | 30 | 10° 11' N | 84° 40' W | 3464 | ?- pre-2012 | 25.1 |
| PISTE-ST.E | GUIANAS | Piste Ste. Elie, French Guiana | Puig and Delobelle, 1988 | 10-50 | 5° N | 53° W | 3238 | 1978-1981 | 26 |
| NOU-GP | GUIANAS | Les Nourages, French Guiana | Chave et al., 2001 | 100 (-411) | 4° 50' N | 50° 42' W | 2757 | 1989-1998 | |
| NOU-PP | GUIANAS | Les Nourages, French Guiana | Chave et al., 2001 | 100 (-411) | 4° 50' N | 50° 42' W | 2757 | 1989-1998 | |
| BISLEY | CARIBB. | Luquillo (Bisley), Puerto Rico | Cusack et al., 2011 | 260 | 18° 20' N | 65° 48' W | 3500 | | |
| LAMBIR | ASIA | Lambir, Sarawak, Borneo | Chave et al., 2008a | 124-209 | 4.1865 | 114.017 | 2921 | | |
| PALANAN | ASIA | Palanan, Philippines | Chave et al., 2008a | 85-140 | 17.0402 | 122.388 | 2607 | | |
| PASOH | ASIA | Pasoh, Malaysia | Chave et al., 2008a | 70-90 | 2.982 | 102.313 | 1973 | | |
| SINHA | ASIA | Sinharaja, Sri Lanka | Chave et al., 2008a | 424-575 | 6.4023 | 80.4023 | 3379 | | |
| MAEKL | ASIA | Mae Klong Stn, Thailand | Takahashi et al., 2012 | 150-350 | 14° 35' N | 98° 52' E | 1650 | pre-1995 | ca. 25 |

[1] Rainfall data from Manaus, in Vieira et al., 2004
[2] Rainfall data from Rio Branco, in Vieira et al., 2004
[3] Rainfall data from Santarem, in Vieira et al., 2004
[4] Elevation data from CTFS website; latitude and longitude from http://daac.ornl.gov/cgi-bin/dsviewer.pl?ds_id=157.

## 5 Conclusions: next steps

A community-consensus catalogue of the benchmark-level field observations directly relevant to C cycling would be a major advance. As we found in this first effort for tropical forests, the development of such catalogues will require the active participation of both field researchers and modelers. Involvement of field researchers with extensive experience in C-cycling studies in the target biome will be critical for identifying reference-level field data. Such an effort will require their extensive



first-hand expertise with field methods and conditions in the target ecosystems, along with broad knowledge of the relevant literature. Field ecologists and modellers are now collaborating at the outset of field experiments to determine the necessary observations for testing ecosystem-level hypotheses embedded in the theoretical components of ESM's. This same interdisciplinary approach is important for identifying appropriate field observations for effective model-data fusion. Given

the increasing use of models as tools for understanding ecosystem processes, a new generation of scientists who can work across empirical and theoretical fields will be key for this effort.

Data catalogues need to be "living" resources, constantly updated as new information comes in and as ecological insights and methods develop in each biome. For the on-going updating, a web-based, moderated system would seem to be the strongest approach. With such a system, field researchers worldwide could actively participate, continuously offering

new field observations for consideration and also correcting or augmenting current entries. Proposed updates, however, should be pre-screened by a team of volunteer researchers and modelers with the relevant expertise.

We have identified here examples of reference-level field observations from old-growth lowland tropical forests. Now what is clearly needed is a much broadened discussion among the wider tropical-research community, both to refine the benchmark criteria for these forests and to contribute observations on a continual basis going forward. A similar parallel

effort is also greatly needed to identify data benchmarks for the highly distinct C-cycling processes taking place in degraded and successional tropical forests, which may account for half or more of the forest area across the tropics (Chazdon, 2014). Yet a different set of benchmarks would be needed to characterize C-cycling in tropical montane forests, an ecologically distinct class of tropical forests.

Our effort here provides a starting point for addressing the modeling community's need for reference-level field

observations from the tropical-forest biome. As is evident from our review, the field data for our target forests are woefully sparse. One critical concern is the total absence of information for some potentially important aspects of C-cycling, such as root exudation and the C exports from plants to their symbionts. More generally, there is a clear need for observations of all aspects of C-cycling to be made at the landscape scale and through time. Such studies need to be made across an expanded set of forests that spans all major tropical regions. These identified needs provide a set of exciting and urgent priorities for

the community of tropical field ecologists. At the same time, our review has provided numerous valuable points of reference from the field studies to date in tropical forests. Following the vision of the ILAMB effort, many aspects of the existing field observations can serve as benchmarks for developing and evaluating the land models with respect to the tropical-forest biome.

*Author contributions.* All authors collaborated in the writing of the paper.

*Acknowledgements.* This work was made possible by the support of the U.S. Geological Survey John Wesley Powell Center for Analysis and Synthesis. The support from the Powell Center included funding the participation of SA. DAC was



supported by US National Science Foundation LTREB grants DEB-1147367 and DEB-1357112. TEW was supported by US Department of Energy, Terrestrial Ecosystem Sciences grant DE-SC0011806 and by the USDA Forest Service International Institute for Tropical Forestry in collaboration with the University of Puerto Rico. XY was supported by the Next Generation Ecosystem Experiments-Tropics and the Biogeochemistry–Climate Feedbacks Scientific Focus Area (BGC Feedbacks SFA)

5    of the US Department of Energy, Office of Science, Office of Biological and Environmental Research. PBR was supported by the US Department of Energy, Office of Science (DE-SC0012677). SR was financially supported by US Department of Energy, Terrestrial Ecosystem Sciences grant DE-SC0011806 and by the US Geological Survey Ecosystems Mission Area. Any use of trade, firm, or product names is for descriptive purposes only and does not imply endorsement by the US Government.  David B. Clark constructively commented on the manuscript.



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
