# Peer review of "Field data to benchmark the carbon-cycle models for tropical forests"

_Biogeosciences, 2017_

## Referee Comment (RC1) · Anonymous Referee #1 · 24 May 2017

Review

The article presents a critical review of the currently available data to test the land-surface and vegetation/carbon dynamic components of Earth System Models in low-land tropical forests. After introducing general criteria for field data to be useful and trustable with regards to benchmark model results (Section 2 and 3), the authors reviewed the available data and associated issues in terms of standing carbon pools (LAI, aboveground and belowground biomass pools, soil organic carbon), ecosystem fluxes (NEE, GPP, respiration, NPP, litterfall) and tree mortality. A couple of final sections mentioned ecosystem C-fluxes sensitivity to climatic trends and availability of local meteorological conditions. The article is very well-written and touches on a sensitive topic of interest for the field experimentalist and modeling communities, bringing upfront many issues that are well-known but not explicitly written in a scientific manuscript. It is

a review, therefore does not introduce any new methods or dataset but it clearly frames the picture of data availability for model comparison in the tropics. Therefore, the article type should be "Reviews and syntheses" and not "Research article". My knowledge of the empirical literature of tropical forest is forcefully partial but I have the impression the authors are including most of what I am aware of in their discussion, even though the reference selection is by far not exhaustive, especially for the fluw-tower data. In any case, I strongly agree with almost everything is written in the manuscript and I have mostly suggestions on points to stress or minor comments as listed below.

What clearly emerges from the manuscript is a gloomy picture of data availability to benchmark model in lowland tropical forests that let me wonder if the correct title should rather be "Field data to benchmark the carbon-cycle models for tropical forests are mostly lacking". Unfortunately, such unsatisfactory amount of data availability corresponds to reality; we do not have almost anything to compare models with at the landscape scale in the tropics. This remarks the challenge of collecting model-meaningful data or add reasonable uncertainty bounds to the current available data in the tropics. It further suggests that some of the model-to-data comparisons carried out in the past for tropical forests might have compared apples to oranges or that the confidence given to certain type of "observed" data (e.g., GPP or aboveground wood production) in previous articles is unjustified. This criticism, which I completely share, is evident throughout the article but it is never really made explicit and probably can be reinforced in a revised version. Additionally, I suggest reinforcing a few other points. (i) It is extremely important collecting sub-daily resolution meteorological data for a number of variables simply to run the models. Practically, for many experimental sites, these data are missing and modelers have to rely on "re-analysis forcing" introducing additional discrepancies in the model-to-data comparison. (ii) Analysis of climatic sensitivities as the one reported in Table 12 (results from Clark et al 2013) are fundamental because they allow to test if the mechanistic nature of the model is able to capture the correct direction and magnitude of a given response and are typically less subjected to local biases than matching a given carbon pool or an uncertain flux. (iii) Given the paucity
of data and their uncertainties, there should be some clear statement about the weakness of automatic calibration or to "force" models to reproduce as close as possible specific observations or set of observations (e.g., eddy covariance fluxes). With all the issues described in this manuscript, it is very unlikely that we are able to constrain several of model parameters using current data. In other words, there should be an effort from modelers in using observations very critically and not blindly and from experimentalists in communicating properly the limitations of measurements and accept model estimates (which are, at least, constrained by mass and energy conservation) critically and not simply as "wrong numbers.

Minor Comments

Page 2, Line 1-4 and elsewhere. I wonder if the article really needs these citations at the beginning of sections. I find a bit unconventional for a review paper and the main message of the citation can be or is already embedded in the main text.

Page 2 and Line 10-13. There are also studies that attributes a large part of the variability of the land CO2 sink to semi-arid ecosystems (Ahlstrom et al 2015). Maybe it can be mentioned together with the role of tropical forest.

Page 5 – Line 5-6. It is a quite trivial statement that the most meaningful variables to compare with are the ones directly observed. At the same time, it is also true that there is some value in comparing model simulations with variables, which are somehow inferred from field observations, even though not directly observed. I would make this a bit more nuanced.

Page 8. Line 14-15. I completely agree on the importance of capturing interannual variability and long-term trends, ultimately this is what we are really interested in, at the same time, it is important to understand the mechanisms leading to these trends/variability, otherwise we risk that models are forced to reproduce something for the wrong reason or through the wrong process.

Page 9. Line 1. "high-resolution local meteorological data" are simply fundamental. For instance, many or almost all the RAINFOR plots will be impossible to simulate properly with models because meteorological data are not available or are not properly released.

Table 1. While there are not estimates for tropical forest, plant-C export to mycorrhizal and root exudates are typically thought to be at the maximum 10-15%. Maybe calling it a "large fraction" is a bit excessive and subjective statement.

Table 7. The estimate for VOC: 10-90 gC/m-2 yr-1 seem to me too large (almost one order of magnitude), when compared to other estimates in the tropics (Kuhm et al 2007) and generally to the expected mass contribution of VOC (Keenan et al 2009)

Page 21. Leaf – litterfall. One point, I would made explicit is that litterfall estimates should be coherent with the leaf turnover rates and the product of average "leaf mass area" [gC/m2 LAI] and LAI [m2 LAI/m2 ground] observed in a given site. My experience, from published observations, not only on the tropics, is that this is rarely the case. Probably, this is the result of the problems you mentioned.

Page 21. Line 26-27, see also Wu et al 2016, for the link between leaf-production and litterfall, even though not completely synchronous.

Page 24. Line 2-10. I would still mention that some observations of fine root production even though sub-optimal is very important, if it is not used blindly in models.

Page 25. Line 12-28. See also Gloor et al 2009.

Page 29. Line 21-22. I do not want to downplay the importance of C-Exudation and export to mycorrhizal, but with uncertainties in NPP and GPP estimates in the order of 20-30%, I would emphasize this aspect in the conclusions not only the missing components, which is likely smaller. In the conclusions, I would also suggest to emphasize more the temporal dynamics of pools and fluxes in line with the Section 3.3 and 4.5.

References

[Figure]

Kuhn, U., et al. (2007), Isoprene and monoterpene fluxes from Central Amazonian rainforest inferred from tower-based and airborne measurements, and implications on the atmospheric chemistry and the local carbon budget, Atmos. Chem. Phys., 7, 2855–2879.

Ahlström A, et al. (2015) Carbon cycle. The dominant role of semi-arid ecosystems in the trend and variability of the land COâĆĆ sink. Science 348(6237):895–899.

Wu, J., et al. (2016). Leaf development and demography explain photosynthetic seasonality in Amazon evergreen forests. Science, 351(6276), 972-976.

Gloor, M., et al. (2009), Does the disturbance hypothesis explain the biomass increase in basin-wide Amazon forest plot data?. Global Change Biology, 15: 2418–2430. doi: 10.1111/j.1365-2486.2009.01891.x

Keenan, T., Niinemets, Ü., Sabate, S., Gracia, C., and Peñuelas, J. (2009) Process based inventory of isoprenoid emissions from European forests: model comparisons, current knowledge and uncertainties, Atmos. Chem. Phys., 9, 4053-4076, doi:10.5194/acp-9-4053-2009.
* * *

---

## Referee Comment (RC2) · Anonymous Referee #2 · 20 Jun 2017

General Comments:

Summary: The goal of the manuscript is to set a benchmark for observational data to be used for the improvement and validation of vegetation carbon cycle models.

The authors report on general challenges that occur in model-data comparison, and on the limitations of data and models at different temporal and spatial scales. These are used to identify criteria for benchmark field data in tropical forests such as landscape-scale sampling and long data series. The authors clarify, in detail, terms of carbon stocks and fluxes, and underline uncertainties that arise from observations. Exemplary for each stock and flux, well-documented field data of tropical forests are identified that fulfill the above-mentioned criteria.

[Figure]

The manuscript summarizes needs of the modeler community and sets a starting point for the development of a benchmark-level field catalogue. The authors conclude that the development of such catalogues requires an active participation of field scientists and modelers and constant maintenance.

Article contribution and overall impact: The manuscript is very well written and gives a good overview and discussion on challenges that are confronted when comparing field data and results of vegetation carbon cycle models.

The manuscript seems rather like a review than a research article as it gathers data from literature and does not introduce new methods or analyses.

As the manuscript is very long, it may benefit from an additional figure in the second part (Chapter 4: Benchmark field data from lowland old-growth tropical forests). You could add a figure in which you demonstrate the different stocks and fluxes and their interactions. Such a figure does not only highlight the very precisely defined terms of different carbon stocks and fluxes, it may also draw attention to potential contributors for the community-consensus catalogue of benchmark-level field inventory.

Overall, the manuscript is an important contribution as it highlights that field researchers and modelers need to work actively together to improve large-scale carbon stock and flux estimates.

Specific comments:

Figure 1: Please add a legend or a description to the figure caption. I assume the different colors refer to the seven climate models and the black line is the mean? What does the grey area represent?

Chapter 2.1 (headers): Why do you need a subsection here, as there is no 2.2?

Page 4, line 16: What do you mean by "hybrid C-cycle"? Estimates that are derived as residuals?

Page 6, line 33-end of paragraph: You here mention "one class" of models. I assume you refer to demographic models such as the ED model and individual-based models such as LPJ-GUESS or SEIB-DGVM. Individual-based DGVMs can also represent within-landscape heterogeneity. In the rest of the manuscript you only refer to "demographic models". This term may not cover the full range of models that can display spatial heterogeneity.

Technical corrections:

Page 11, line 2: "underestimated" instead of "underestimate".

Page 18, line 25: typo in R_eco

Page 22, line 22: "measurements" instead of "measurement".

---

## Author Comment (AC1) · 22 Jul 2017

Authors' responses to comments of Referee #1 (21 July 2017):

We much appreciate this referee's constructive and thoughtful comments. Below we have pasted in the entire review, and we have inserted our responses to the suggestions and questions (indicated by bracketing stars).
Review The article presents a critical review of the currently available data to test the land- surface and vegetation/carbon dynamic components of Earth System Models in lowland tropical forests. After introducing general criteria for field data to be useful

and trustable with regards to benchmark model results (Section 2 and 3), the authors reviewed the available data and associated issues in terms of standing carbon pools (LAI, aboveground and belowground biomass pools, soil organic carbon), ecosystem fluxes (NEE, GPP, respiration, NPP, litterfall) and tree mortality. A couple of final sections mentioned ecosystem C-fluxes sensitivity to climatic trends and availability of local meteorological conditions. The article is very well-written and touches on a sensitive topic of interest for the field experimentalist and modeling communities, bringing upfront many issues that are well-known but not explicitly written in a scientific manuscript. It is a review, therefore does not introduce any new methods or dataset but it clearly frames the picture of data availability for model comparison in the tropics. Therefore, the article type should be "Reviews and syntheses" and not "Research article".

\*\*We agree.\*\*

My knowledge of the empirical literature of tropical forest is forcefully partial but I have the impression the authors are including most of what I am aware of in their discussion, even though the reference selection is by far not exhaustive, especially for the fluw-tower data. In any case, I strongly agree with almost everything is written in the manuscript and I have mostly suggestions on points to stress or minor comments as listed below. What clearly emerges from the manuscript is a gloomy picture of data availability to benchmark model in lowland tropical forests that let me wonder if the correct title should rather be "Field data to benchmark the carbon-cycle models for tropical forests are mostly lacking". Unfortunately, such unsatisfactory amount of data availability corresponds to reality; we do not have almost anything to compare models with at the landscape scale in the tropics. This remarks the challenge of collecting model-meaningful data or add reasonable uncertainty bounds to the current available data in the tropics.

It further suggests that some of the model-to-data comparisons carried out in the past for tropical forests might have compared apples to oranges or that the confidence

given to certain type of "observed" data (e.g., GPP or aboveground wood production) in previous articles is unjustified. This criticism, which I completely share, is evident throughout the article but it is never really made explicit and probably can be reinforced in a revised version.

\*\*We are happy to add explicit statements to reinforce the points in the paper about this problem of inappropriate comparisons in past model-to-data evaluation studies.\*\*

Additionally, I suggest reinforcing a few other points.

(i) It is extremely important collecting sub-daily resolution meteorological data for a number of variables simply to run the models. Practically, for many experimental sites, these data are missing and modelers have to rely on "re-analysis forcing" introducing additional discrepancies in the model-to-data comparison.

\*\*We very much agree with this point, and would be happy to add a strong statement about it to the conclusions section.\*\*

(ii) Analysis of climatic sensitivities as the one reported in Table 12 (results from Clark et al 2013) are fundamental because they allow to test if the mechanistic nature of the model is able to capture the correct direction and magnitude of a given response and are typically less subjected to local biases than matching a given carbon pool or an uncertain flux.

\*\*We agree, and would be happy to add a strong statement about it in the conclusions section and abstract.\*\*

(iii) Given the paucity of data and their uncertainties, there should be some clear statement about the weakness of automatic calibration or to "force" models to reproduce as close as possible specific observations or set of observations (e.g., eddy covariance fluxes). With all the issues described in this manuscript, it is very unlikely that we are able to constrain several of model parameters using current data. In other words, there should be an effort from modelers in using observations very critically and not blindly

and from experimentalists in communicating properly the limitations of measurements and accept model estimates (which are, at least, constrained by mass and energy conservation) critically and not simply as "wrong numbers.

\*\*We agree. We will edit as appropriate to reinforce this point.\*\*

Minor Comments

Page 2, Line 1-4 and elsewhere. I wonder if the article really needs these citations at the beginning of sections. I find a bit unconventional for a review paper and the main message of the citation can be or is already embedded in the main text.

\*\*We feel the value of including these quotations lies in explicitly presenting current thinking of the research community with respect to the central points of the adjacent section. Just citing those papers in the text would not provide this context so clearly, especially for those readers less familiar with those publications.\*\*

Page 2 and Line 10-13. There are also studies that attributes a large part of the variability of the land $CO_2$ sink to semi-arid ecosystems (Ahlstrom et al 2015). Maybe it can be mentioned together with the role of tropical forest.

\*\*Lines 10-13 address specifically the negative relation between tropical temperatures and the atmospherically-inferred C balance of the land tropics in toto (i.e., the joint effect of tropical forests, tropical semi-arid systems, agricultural systems, etc.). We then turn to questions directly related to tropical forests, the focus of the paper. We feel that remaining focused on tropical forests would be more effective than discussing here the different conclusions of recent studies inferring the contribution of semi-arid systems to the land C balance.\*\*

Page 5 – Line 5-6. It is a quite trivial statement that the most meaningful variables to compare with are the ones directly observed. At the same time, it is also true that there is some value in comparing model simulations with variables, which are somehow inferred from field observations, even though not directly observed. I would make this

a bit more nuanced.

**We agree that partially-extrapolated or otherwise inferred indirect estimates can have heuristic value, and we will attempt to express this more explicitly; however, the focus of this paper is benchmark-level field data, those providing the most solid possible standards for guiding the models.**

Page 8. Line 14-15. I completely agree on the importance of capturing interannual variability and long-term trends, ultimately this is what we are really interested in, at the same time, it is important to understand the mechanisms leading to these trends/variability, otherwise we risk that models are forced to reproduce something for the wrong reason or through the wrong process.

**We agree.**

Page 9. Line 1. "high-resolution local meteorological data" are simply fundamental. For instance, many or almost all the RAINFOR plots will be impossible to simulate properly with models because meteorological data are not available or are not properly released.

**We agree (and, as noted earlier, we would be happy to reinforce this point in the conclusions section).**

Table 1. While there are not estimates for tropical forest, plant-C export to mycorrhizal and root exudates are typically thought to be at the maximum 10-15%. Maybe calling it a "large fraction" is a bit excessive and subjective statement.

**We would be happy to change the comment for each of these C fluxes in Table 1 from "Unquantified in tropical forest; possibly a large and increasing fraction of NPP" to "Unquantified; possibly a non-trivial and/or increasing NPP fraction".**

Table 7. The estimate for VOC: 10-90 gC/m-2 yr-1 seem to me too large (almost one order of magnitude), when compared to other estimates in the tropics (Kuhm et al 2007) and generally to the expected mass contribution of VOC (Keenan et al 2009)

**Per Guenther et al.'s 1995 (JGR-Atmospheres, 100:8873-8892) model of tropical rain forest total VOC emissions (isoprene, monoterpenes, other reactive VOC [ORVOC], and other VOC [OVOC]), annual emissions of all these classes of VOC combined are estimated at 31 g C m-2 (from their Table 1). They state that "...estimates of annual VOC fluxes from tropical forests are as high as 75 g C m-2." They also conclude the uncertainties exceed a factor of 3. We base our guesstimates in that table on these numbers, which include additional VOC's (beyond isoprenes and monoterpenes).**

Page 21. Leaf – litterfall. One point, I would made explicit is that litterfall estimates should be coherent with the leaf turnover rates and the product of average "leaf mass area" [gC/m2 LAI] and LAI [m2 LAI/m2 ground] observed in a given site. My experience, from published observations, not only on the tropics, is that this is rarely the case. Probably, this is the result of the problems you mentioned.

**Such a cross check would indeed be excellent in any system where the two data-types are available. Unfortunately, we know of no tropical forest with direct observations for estimating the whole-forest leaf turnover rate (an appropriately-weighted composite of the mean leaf turnover rates for the understory, mid-storey, and canopy). Leaf longevities in the understory of tropical moist/wet forests have been found to exceed several years, while leaf longevities in the canopy can vary from more than 1 yr to several months, depending on the tree/liana species.**

Page 21. Line 26-27, see also Wu et al 2016, for the link between leaf-production and litterfall, even though not completely synchronous.

**That very interesting study used ground-based observations through time of the "% of individuals with leaves < 4 mo old", which varied between ca. 25% and 50%, as a surrogate for leaf production. Although providing a useful seasonal index related to leaf phenology, this metric would not enable quantification of forest-level leaf-production or its exact phenology through time.**

Page 24. Line 2-10. I would still mention that some observations of fine root production even though sub-optimal is very important, if it is not used blindly in models.

\*\*We agree. As we state at the outset of the fine-root production section, when observations are highly replicated at the landscape scale, they provide a useful lower bound (lower bound, because they are confined to the surface soil and also involved low-bias from the methods issues). We will reinforce this point in the text.\*\*

Page 25. Line 12-28. See also Gloor et al 2009.

\*\*We agree this reference is a good addition to the citations here.\*\*

Page 29. Line 21-22. I do not want to downplay the importance of C-Exudation and export to mycorrhizal, but with uncertainties in NPP and GPP estimates in the order of 20-30%, I would emphasize this aspect in the conclusions not only the missing components, which is likely smaller. In the conclusions, I would also suggest to emphasize more the temporal dynamics of pools and fluxes in line with the Section 3.3 and 4.5.

\*\*We agree that adding both these points would strengthen the final conclusions.\*\*

(end of author responses to comments by Referee #1)

---

## Author Comment (AC2) · 22 Jul 2017

Authors' responses to comments of Referee #2 (21 July 2017):

We much appreciate this referee's constructive and thoughtful comments. Below we have pasted in the entire review, and we have inserted our responses to the suggestions and questions (indicated by bracketing stars).
Review General Comments: Summary: The goal of the manuscript is to set a benchmark for observational data to be used for the improvement and validation of vegetation carbon cycle models.

The authors report on general challenges that occur in model-data comparison, and on the limitations of data and models at different temporal and spatial scales. These are used to identify criteria for benchmark field data in tropical forests such as landscape scale sampling and long data series. The authors clarify, in detail, terms of carbon stocks and fluxes, and underline uncertainties that arise from observations. Exemplary for each stock and flux, well-documented field data of tropical forests are identified that fulfill the above-mentioned criteria.

The manuscript summarizes needs of the modeler community and sets a starting point for the development of a benchmark-level field catalogue. The authors conclude that the development of such catalogues requires an active participation of field scientists and modelers and constant maintenance.

Article contribution and overall impact: The manuscript is very well written and gives a good overview and discussion on challenges that are confronted when comparing field data and results of vegetation carbon cycle models.

The manuscript seems rather like a review than a research article as it gathers data from literature and does not introduce new methods or analyses.

**We agree with you and Referee 1 that the paper is of the type "Reviews and Syntheses."**

As the manuscript is very long, it may benefit from an additional figure in the second part (Chapter 4: Benchmark field data from lowland old-growth tropical forests). You could add a figure in which you demonstrate the different stocks and fluxes and their interactions. Such a figure does not only highlight the very precisely defined terms of different carbon stocks and fluxes, it may also draw attention to potential contributors for the community-consensus catalogue of benchmark-level field inventory.

**We considered such a figure but have opted not to attempt this, given the large uncertainties about the magnitudes of many C stocks and fluxes in tropical forests.**

Overall, the manuscript is an important contribution as it highlights that field researchers and modelers need to work actively together to improve large-scale carbon stock and flux estimates.

**We strongly concur with the latter point.**

Specific comments:

Figure 1: Please add a legend or a description to the figure caption. I assume the different colors refer to the seven climate models and the black line is the mean? What does the grey area represent?

**Thank you for noting these omissions. We will add these and additional details to the legend (see below here), and we will also add to the figure the inadvertently omitted color key to the individual models. [ Revised figure legend: Figure 1. Divergent projections (colored lines) of the changes in tropical Net Ecosystem Production through this century from seven of the CMIP5 climate models. The key identifies the models. Dashed lines - models that include coupled carbon–nitrogen (C-N) biogeochemistry; solid lines - models lacking explicit nutrient cycling. The ensemble mean is indicated by the heavy black line, and gray shading indicates the range of one standard deviation ($1\delta$) in climate model variability (adopted with permission from Cavaleri et al., 2015 [© 2015 John Wiley & Sons Ltd]). ]**

Chapter 2.1 (headers): Why do you need a subsection here, as there is no 2.2?

**We considered the header valuable for emphasizing the "apples to apples" point. If desired, however, we could drop this section header and re-work the first sentence of the following paragraph to underline the point.**

Page 4, line 16: What do you mean by "hybrid C-cycle"? Estimates that are derived as residuals?

**We used the term "hybrid" to refer to a C-cycle estimate that is based partly on direct measurements and partly on extrapolation. This point is further discussed, with examples, in the following section. We agree that the text of this current section can be improved by more clearly defining the term at the outset.**

Page 6, line 33-end of paragraph: You here mention "one class" of models. I assume you refer to demographic models such as the ED model and individual-based models such as LPJ-GUESS or SEIB-DGVM. Individual-based DGVMs can also represent within-landscape heterogeneity. In the rest of the manuscript you only refer to "demographic models". This term may not cover the full range of models that can display spatial heterogeneity.

**We agree and will edit here (and elsewhere in the paper, as needed) to clarify that there are multiple model-types that explicitly represent spatial heterogeneity within a landscape.**

Technical corrections:

Page 11, line 2: "underestimated" instead of "underestimate".

**We believe "underestimate" is the correct form, since the sentence is "Field observations .... typically underestimate [LAI]."**

Page 18, line 25: typo in R_eco

**We don't see the typo here (?).**

Page 22, line 22: "measurements" instead of "measurement".

**We will correct to plural.**

(end of author responses to comments by Referee #2)

---

## Author Response (AR1)

Dear Dr. Rammig, Editor,

(19 August 2017)

We are submitting here our revised manuscript, "Field data to benchmark the carbon-cycle models for tropical forests; D.A. Clark et al." for further consideration by Biogeosciences.

We would appreciate editorial transfer of our paper from the category "Research Article" to the category "Reviews and Syntheses", as recommended by the two referees.

Here we detail the changes made to the original submission in response to the referees' comments (in our prior posted response to each referee we indicated the reasons for not making some recommended changes/additions). This listing follows the sequence of each referee's comments:

Referee #1:

- (paragraph starting on p.4, line 13) This paragraph was modified in response to both referees' comments by: a) removing the section header 2.1 and instead incorporating the section title in a revised first sentence; b) reinforcing the discussion of inappropriate comparisons of model results to C-cycle estimates derived only partially from field observations, including adding a second example from two high-profile studies; and c) defining such "hybrid estimates" here where they are first mentioned.

- (lines 3-4, p. 27; lines 25-26, p. 29).  We added text about the importance of sub-daily resolution met data to the section on meteorology and to the Conclusions section (Ref. #1).

- (lines 26-28, p. 29).  We added a sentence about the importance and rarity of analyses of the climatic/CO2 sensitivities of tropical-forest C-cycling to the Conclusions section (Ref. #1).

- (abstract, line 26).  We added analyses of climatic/$CO_2$ and of long-term trends to the list of needed benchmark data types in the abstract (Ref. #1).

- (p. 5, lines 15-17). We added brief text recognizing the importance of field researchers communicating the underlying methods and limitations associated with their observations and of the modelers critically evaluating the observations in this light before using them in model-data exercises (Ref. #1).

- Table 1 (p. 10). We changed the comments in this table for the C export to mycorrhizae/ nodules and for the C in root exudates from "Unquantified in tropical forest; possibly a large and increasing fraction of NPP" to "Unquantified; possibly a non-trivial and/or increasing NPP fraction".

- (p. 19, lines 28-32) We added and briefly discuss relevant references for VOC production by tropical forests (comment by Ref. # 1 on Table 7).

- (p.23, lines 32-33). We revised the initial, summary sentence on fine-root production to highlight that landscape-scale field estimates serve as a lower bound.

- (p. 25, line 16) We added the Gloor et al. 2009 reference (Ref. #1).

- (p. 29, lines 19-32). We re-wrote the concluding paragraph of the conclusions to better highlight the issues of overall uncertainty and the need for data through time to monitor dynamics and trends.

Referee #2:

- (Figure 1) We added to the figure the color key that identifies the results by model, and we expanded the legend to include the missing explanations about the figure, as identified by the referee.

- (paragraph starting on p.4, line 13) This paragraph was modified in response to both referees' comments by: a) removing the section header 2.1 and instead incorporating the section title in a revised first sentence; b) reinforcing the discussion of inappropriate comparisons of model results to C-cycle estimates derived only partially from field observations, including adding a second example from two high-profile studies; and c) defining such "hybrid estimates" here where they are first mentioned.

- (p. 7, paragraph starting on line 4; p. 12, lines 24-25).  We re-wrote these sections of text to correct the omission of the individual-based models from the discussion of those models that explicitly represent the spatial heterogeneity within landscapes.

- (p. 22, line 23).  We corrected "measurement" to "measurements."

We thank both referees for their detailed reviews and constructive comments.  We believe the paper was significantly strengthened by this input.

[revised manuscript text omitted]